

# A review of the diagnosis and geographical distribution of the recently described flea toad *Brachycephalus sulfuratus* in relation to *B. hermogenesi* (Anura: Brachycephalidae)

Marcos R. Bornschein[1,2], Luiz Fernando Ribeiro[2], Larissa Teixeira[1], Ricardo Belmonte-Lopes[2], Leonardo Amaral de Moraes[3], Leandro Corrêa[2], Giovanni Nachtigall Maurício[3], Júnior Nadaline[2,4] and Marcio R. Pie[2,4]

[1] Departamento de Ciências Biológicas e Ambientais, Universidade Estadual Paulista, São Vicente, São Paulo, Brazil
[2] Mater Natura - Instituto de Estudos Ambientais, Curitiba, Paraná, Brazil
[3] Programa de Pós-Graduação em Biologia Animal, Universidade Federal de Pelotas, Pelotas, Rio Grande do Sul, Brazil
[4] Departamento de Zoologia, Universidade Federal do Paraná, Curitiba, Paraná, Brazil

Corresponding author
Marcos R. Bornschein,
marcos.bornschein@unesp.br

## ABSTRACT

**Background:** The flea toad *Brachycephalus sulfuratus* was recently described from southeastern and southern Brazil. In its description, the authors overlooked previous records of flea toads that had been identified as "*Brachycephalus* sp. nov." and *B. hermogenesi* occurring in the same regions, which could suggest the possibility of up to three flea toads coexisting in southern Brazil. In addition, *B. sulfuratus* is characterized by substantial phenotypic variability, to an extent that compromises its current diagnosis with respect to its congener *B. hermogenesi*. Therefore, the current state-of-affairs regarding the geographical distribution of these two species and the identification of previously known populations is hitherto uncertain. Our goals are to reassess previous records of flea toads attributable to *B. hermogenesi*, *B. sulfuratus* and "*Brachycephalus* sp. nov.", considering the description of *B. sulfuratus*, and to review the diagnosis of *B. sulfuratus*.

**Methods:** A critical analysis of the species identity of flea toad specimens attributable to *B. hermogenesi*, *B. sulfuratus*, or to a potentially undescribed species from southeastern and southern Brazil was based either on the analysis of morphology or on their advertisement calls. These analyses include our independent examinations of specimens and, when not possible, examinations of published descriptions. To allow for a consistent comparison of advertisement calls between *B. hermogenesi* and *B. sulfuratus*, we made recordings of both species, including in the type locality of the former.

**Results:** We found that morphological and call characters originally proposed as diagnostic for *B. sulfuratus* in relation to *B. hermogenesi* vary intraspecifically. Live individuals with ventral yellow spots correspond to *B. sulfuratus*; individuals without yellow spots can be either *B. sulfuratus* or *B. hermogenesi*. In preservative, they are indistinguishable. Previous records of *Brachycephalus* sp. nov. correspond to
*B. sulfuratus.* We propose that the reduced number of notes per call and the presence of only isolated notes in the call of *B. sulfuratus*, as opposed to a high number of notes per call with isolated notes and note groups in the call of *B. hermogenesi*, as the only diagnostic characters between them. Regarding their distributions and based in our assessment, only *B. sulfuratus* occurs in southern Brazil, without any overlap with *B. hermogenesi*. There is a narrow gap between the distributions of these species around the southeast of the city of São Paulo. Our revision also revealed that some records previously attributed to *B. hermogenesi* in Rio de Janeiro and north São Paulo represent a distinct, unidentified flea toad that is not *B. sulfuratus*. Both species occur side by side in Corcovado, São Paulo, a locality from where five paratypes of *B. hermogenesi* were obtained. Biogeographic events that might have led to vicariance between *B. hermogenesi* and *B. sulfuratus* are discussed.

# INTRODUCTION

The genus *Brachycephalus* Fitzinger, 1826 includes 36 small diurnal anuran species that live in the leaf litter across the Brazilian Atlantic Rainforest (*Bornschein, Pie & Teixeira, 2019*). Most species present small geographic distributions, restricted to one or a few adjacent mountaintops (*Pie et al., 2013*; *Bornschein et al., 2016a*; *Bornschein, Pie & Teixeira, 2019*). *Brachycephalus* has been divided in three phenetic groups, based on body shape and presence/absence of dermal co-ossification (*Ribeiro et al., 2015*), and presence/ absence of *linea masculinea* (*Pie et al., 2018b*): the *B. ephippium* group, with 12 species distributed from Espírito Santo and Minas Gerais south to São Paulo, southeastern Brazil (*Bornschein, Pie & Teixeira, 2019*); the *B. pernix* group, with 19 species distributed in Paraná and Santa Catarina, southern Brazil (*Bornschein, Pie & Teixeira, 2019*); and the *B. didactylus* group, with four species commonly known as flea toads and distributed throughout much the Atlantic Forest of Brazil, from Bahia to Santa Catarina, northeastern, southeastern, and southern Brazil (*Bornschein, Pie & Teixeira, 2019*). Members of the *B. didactylus* species group (sensu *Ribeiro et al., 2015*; *Pie et al., 2018a*; *Bornschein, Pie & Teixeira, 2019*) are distinguished by their leptodactyliform body shape and the absence of dermal ossification and absence of *linea masculinea*. The *B. ephippium* species group includes species with bufoniform body shape, presence of dermal ossification and absence of *linea masculinea*, and, finally, the *B. pernix* species group includes species equally with bufoniform body shape but without dermal ossification and with *linea masculinea* (*Ribeiro et al., 2015*; *Pie et al., 2018a*).

The first described flea toad species was *B. didactylus*, in 1971 (*Izecksohn, 1971*) as the only member of a new genus, *Psyllophryne*. The second flea toad species, *B. hermogenesi*, was described nearly three decades later, in 1998 (*Giaretta & Sawaya, 1998*), at the time as the second species of the genus *Psyllophryne*. This genus was then synonymized in favor of *Brachycephalus* when it was discovered that this genus also had an omosternum,

whose presence until then exclusive in *Psyllophryne* diagnosed that genus in relation to *Brachycephalus* (*Kaplan, 2002*). Recently, other two flea toads were described, namely *B. pulex* (*Napoli et al., 2011*) and *B. sulfuratus* (*Condez et al., 2016*). Only recently have flea toads been recorded in southern Brazil. The first records were of *B. hermogenesi* to the Reserva Particular do Patrimônio Natural Salto Morato (RPPNSM), municipality of Guaraqueçaba, in the northern coast of Paraná (*Pereira et al., 2010*; *Santos-Pereira et al., 2011*) and at Colônia Castelhanos, municipality of Guaratuba, in southern Paraná, initially as "*Brachycephalus* aff. *hermogenesi*" (*Cunha, Oliveira & Hartmann, 2010*) and later as "*B. hermogenesi*" (*Oliveira et al., 2011*). Shortly thereafter, *Pie et al. (2013)* published 14 localities of a flea toad identified as "*Brachycephalus* sp. nov. 1", from Paraná and Santa Catarina. These authors also reidentified the record from Colônia Castelhanos as "*Brachycephalus* sp. nov. 1". Occurrences from RPPNSM of *Pereira et al. (2010)* and *Santos-Pereira et al. (2011)* were overlooked by *Pie et al. (2013)*. Later, *Bornschein et al. (2016a)* compiled 18 localities of a flea toad as *Brachycephalus* sp. 1., including the 14 localities of *Pie et al. (2013)* treated as "*Brachycephalus* sp. nov. 1". *Bornschein et al. (2016a)* also reidentified previous records of the flea toad of the RPPNSM and Colônia Castelhanos as *Brachycephalus* sp. 1.

After these discoveries, the flea toad *B. sulfuratus* was described in 2016 based on a series of 28 specimens distributed from southern São Paulo to northern Santa Catarina (*Condez et al., 2016*). However. these authors did not take into account the information available in *Pie et al. (2013)* and *Bornschein et al. (2016a)*. Rather, *Condez et al. (2016)* only considered the presence of the flea toad *B. hermogenesi* in Paraná, based on *Oliveira et al. (2011)*. However, the voucher specimen of *Oliveira et al. (2011)*, a single specimen deposited in the Museu de História Natural, Universidade Estadual de Campinas, Campinas (ZUEC 16602), was reidentified by *Condez et al. (2016)* as *B. sulfuratus*, whereas the remaining records of *B. hermogenesi* in Paraná, from *Pereira et al. (2010)* and *Santos-Pereira et al. (2011)*, were not considered by *Condez et al. (2016)*.

The absence of a nomenclatural review of records of flea toads in southern Brazil can be evidenced by the fact that a single location in Santa Catarina, called Castelo dos Bugres, was recorded as harboring specimens identified as "*Brachycephalus* sp. nov. 1" (*Pie et al., 2013*), or *Brachycephalus* sp. 1. (*Bornschein et al., 2016a*) and *B. sulfuratus* (*Condez et al. (2016)*. No analysis has been carried out to ensure that the unidentified species represents *B. sulfuratus*, so that the uncertainty in the identification of some important occurrence records seems to indicate three possible scenarios. First, one could envision that potentially there are three similar species of flea toads in Paraná and Santa Catarina, southern Brazil, namely *B. hermogenesi* (*Pereira et al., 2010*; *Santos-Pereira et al., 2011*), *Brachycephalus* sp. (*Pie et al., 2013*; *Bornschein et al., 2016a*) and *B. sulfuratus* (*Condez et al., 2016*). Second, records of *B. hermogenesi* in southern Brazil could be erroneous, given that some of these records (*Cunha, Oliveira & Hartmann, 2010*; *Oliveira et al., 2011*) were assigned to *B. sulfuratus* or "*Brachycephalus* sp. nov." (*Pie et al., 2013*; *Condez et al., 2016*), leading to an expectation that two species might occur in these regions (*B. sulfuratus* and *Brachycephalus* sp.). Third, if the unidentified species of

*Pie et al. (2013)* and *Bornschein et al. (2016a)* is conspecific of *B. sulfuratus*, there could be a single species of flea toad in southern Brazil (*B. sulfuratus*).

Recently, *Bornschein, Pie & Teixeira (2019)* reviewed the available occurrence records of flea toads from southeastern and southern Brazil and reverted most of the records of "*Brachycephalus* sp. nov. 1" (*Pie et al., 2013*), "*Brachycephalus* sp. 1" (*Bornschein et al., 2016a*), and *B. hermogenesi* from southern Brazil (*Pereira et al., 2010*; *Santos-Pereira et al., 2011*, *2016*) in favor of *B. sulfuratus*. Some records that could not be adequately reassessed by *Bornschein, Pie & Teixeira (2019)* were reverted to "*Brachycephalus* sp. cf. *B. sulfuratus*", including the records of *B. hermogenesi* from *Cunha, Oliveira & Hartmann (2010)* and *Oliveira et al. (2011)*. *Bornschein, Pie & Teixeira (2019)* disregarded the possibility of a third unnamed species of flea toad in southern Brazil, but one question remains: the proper identification of *B. sulfuratus* and *B. hermogenesi*. In this sense, the identification criteria used by *Bornschein, Pie & Teixeira (2019)* to reevaluate the records of flea toads were not indicated. In addition, there may still be uncertainty in the identification of flea toads by other authors, as records of *B. hermogenesi* in southern Brazil continue to be published (*Santos-Pereira et al., 2016*; *Santos-Pereira, Pombal & Rocha, 2018*; *Leivas et al., 2018*). Given this uncertainty, the aim of this study is to reanalyze the diagnostic morphological characters used to distinguish *B. sulfuratus* from *B. hermogenesi* and redefine their geographical distributions and distributional limits.

## MATERIALS AND METHODS

The critical analysis of the species identity of specimens attributable to *Brachycephalus hermogenesi*, *B. sulfuratus*, and to a potentially undescribed flea toad from southeastern and southern Brazil provided in our study was based either on the analysis of their morphology or on their advertisement calls. We looked for records in museum specimens, in acoustic collections, and in the literature. The analyzed museum collections include Museu de História Natural Capão da Imbuia (MHNCI), Curitiba, Paraná, Brazil, Coleção Herpetológica do Departamento de Zoologia (DZUP), Universidade Federal do Paraná, Curitiba, Paraná, Brazil, and Museu de História Natural (ZUEC), Universidade Estadual de Campinas, Campinas, São Paulo, Brazil. The sound collection analyzed include MHCNI, Xeno-Canto sound collection (www.xeno-canto.org), and Fonoteca Neotropical Jacques Vielliard (FNJV; https://www2.ib.unicamp.br/fnjv/).

The analyses began by the assessment of the original diagnosis of *B. sulfuratus* (*Condez et al., 2016*). We looked for the proposed diagnostic characters in museum specimens, calls, sources provided in the literature, and our own photographs of live specimens. Given that this procedure uncovered ambiguity in the proposed diagnostic characters to separate *B. sulfuratus* from *B. hermogenesi*, we sought for new characters that could be useful to distinguish them. New distinctive characters were then erected as diagnostic characters, acting in accordance of the Recommendation 13A of the International Code of Zoological Nomenclature (http://www.iczn.org/).

When comparing the calls between *B. sulfuratus* and *B. hermogenesi*, we noticed that the calls of *B. hermogenesi* described by *Verdade et al. (2008)* were from a site 112 km distant in

a straight line from the type locality of this species (*Giaretta & Sawaya, 1998*). As this distance is considerable in relation to distances between other species of the genus (*Pie et al., 2013*; *Bornschein et al., 2016a*), we made additional recordings in the type localities of *B. hermogenesi* (Núcleo Picinguaba and Corcovado; *Giaretta & Sawaya (1998)*) and in the locality where *Verdade et al. (2008)* described the calls of this species (Estação Biológica de Boracéia), as well as in other locations of records of *B. hermogenesi* (e.g., Parque Natural Municipal Nascentes de Paranapiacaba; *Verdade, Rodrigues & Pavan, 2009*).

Our recordings, deposited in the MHNCI, were made using analogical (Sony TCM–5000EV) and digital (Marantz PMD660, Sony PCM–D50 and PCM–M10 and Tascam DR44-WL) devices, with Sennheiser ME 66 and ME 67 microphones. Analogical recordings were digitized at 44.1 kHz and 16 bit using Raven Pro 1.4 (Cornell Lab of Ornithology, Ithaca, NY, USA). Digital recordings were made equally with sampling frequency rate of 44.1 kHz and 16-bit resolution. We analyzed calls under note-centered approach (*Köhler et al., 2017*), as *Bornschein et al. (2018*, *2019)* and *Pie et al. (2018b)*. The definition of call used by *Condez et al. (2016)* is the one defined by *Köhler et al. (2017)* as note-centered approach, in which several notes emitted continuously over a period represent the call of the species, in contrast to the call-centered approach, in which each note represents a call. Remaining call terminology used were those of *Bornschein et al. (2018)*. Spectrograms were produced using Seewave package, version 2.1.6 (*Sueur, Aubin & Simonis, 2008*), in R. 4.0.3 (*R Core Team, 2018*). We made adjustments in contrast and brightness with the intention of lightening the images and best highlighting the pulses. We chose not to noise-filter the spectrograms to avoid eliminating sound characters.

We also included unpublished records in an analysis of *B. sulfuratus* and *B. hermogenesi*, vouchered with specimens collected and deposited in the MHNCI. Collection permits were issued by ICMBIO (10.500, 22470–2/1911426 and 55918–1). Geographical coordinates are based on the WGS84 datum. Elevations for literature records and author's records were obtained from Google Earth, after plotting the location point (*Bornschein et al., 2016a*).

Finally, we generated a phylogenetic tree based on a concatenated dataset of all mitochondrial 12S and 16S mitochondrial loci available on GenBank for specimens of the *B. didactylus* species group (Table S1). Sequences were aligned using MAFFT (*Katoh et al., 2002*) and analyzed under a single GTRGAMMA model in RAxML 8.2.12 (*Stamatakis, 2014*). Support values were obtained by bootstrapping using the automatic halting option. The final tree was rooted by its midpoint. Whenever possible, the corresponding localities available on their GenBank records were standardized based on the toponyms indicated in Table 1.

## RESULTS

Our list of specimens and calls analyzed of *B. sulfuratus* and *B. hermogenesi*, per locality, is provided in Table 1 and Appendix 1.

**Table 1 Current identification of records of flea toads at some point identified as *Brachycephalus sulfuratus*, *B. hermogenesi*, and as an unidentified related species, southeastern and southern Brazil.**

| Species | Locality and state | Geographical coordinates and altitude | Previous identification | Voucher | Our analysis of the record |
|---|---|---|---|---|---|
| *B. sulfuratus* | | | | | |
| *B. sulfuratus* | Bairro Rio Vermelho, municipality of Barra do Turvo, São Paulo | 24°59′25″S, 48°32′26″W; 790 m a.s.l. | — | Specimen | Specimen examined (MHNCI 11584) |
| *B. sulfuratus* | Base of the Serra Água Limpa, municipality of Apiaí, São Paulo | 24°28′52″S, 48°47′12″W; 920 m a.s.l. | Without species identification: *Firkowski et al. (2016)*; *Brachycephalus* sp. 1: *Bornschein et al. (2016a)*; *B. sulfuratus*: *Bornschein et al. (2016b)*, *Ribeiro et al. (2017)*, *Pie et al. (2018b)* | Specimen, calls, and genetic sequence on GenBank | Specimen (MHNCI 11583; Fig. 1F) and calls examined (MHNCI 129; Fig. 3B); KX198030.1 analyzed sequence (Fig. 7) |
| *B. sulfuratus* | Biquinha, municipality of Juquiá, São Paulo | 24°17′43″S, 47°36′26″W; 40 m a.s.l. | *B. sulfuratus*: *Bornschein, Pie & Teixeira, 2019* | Calls | Calls examined (MHNCI 128) |
| *B. sulfuratus* | Braço do Norte, municipality of Itapoá, Santa Catarina | 26°07′29″S, 48°43′48″W; 240 m a.s.l. | *B. sulfuratus*: *Monteiro et al. (2018b)* | Specimen and genetic sequence on GenBank | MG889430.1 analyzed sequence (Fig. 7) |
| *B. sulfuratus* | Caratuval, near the Parque Estadual das Lauráceas, municipality of Adrianópolis, Paraná | 24°51′17″S, 48°43′43″W; 900 m a.s.l. | Without species identification: *Firkowski et al. (2016)*; *Brachycephalus* sp. nov. 1: *Pie et al. (2013)*; *Brachycephalus* sp. 1: *Bornschein et al. (2016a)*; *B. sulfuratus*: *Bornschein et al. (2016b)*, *Ribeiro et al. (2017)*, *Pie et al. (2018b)* | Specimen, calls, and genetic sequence on GenBank | Specimen (MHNCI 11571; Fig. 1B) and calls examined (MHNCI 131); KX198031.1 analyzed sequence (Fig. 7) |
| *B. sulfuratus* | Caratuval, Parque Estadual das Lauráceas, municipality of Adrianópolis, Paraná | 24°51′14″S, 48°42′01″W; 890 m a.s.l. | *Brachycephalus* sp. nov. 1: *Pie et al. (2013)*; *Brachycephalus* sp. 1: *Bornschein et al. (2016a)* | Calls | Calls examined (MHNCI 132) |
| *B. sulfuratus* | Castelo dos Bugres, municipality of Joinville, Paraná | 26°13′47″S, 49°03′20″W; 790–860 m a.s.l. | *Brachycephalus* sp. nov. 1: *Pie et al. (2013)*; *Brachycephalus* sp. 1: *Bornschein et al. (2016a)*; *B. sulfuratus*: *Condez et al. (2016)*, *Monteiro et al. (2018b)* | Specimen, calls, and genetic sequence on GenBank | MK697439.1, MK697487.1, KU321533.1, and MK697390.1 analyzed sequence (Fig. 7) |
| *B. sulfuratus* | Centro de Estudos e Pesquisas Ambientais da Univille, Vila da Glória, Distrito do Saí, municipality of São Francisco do Sul, Santa Catarina | 26°13′39″S, 48°41′31″W; 125 m a.s.l. | *B. sulfuratus*: *Condez et al. (2016)* | Specimen, calls, and genetics | — |
| *B. sulfuratus* | Corvo, municipality of Quatro Barras, Paraná | 25°20′17″S, 48°54′56″W; 930 m a.s.l. | Without species identification: *Firkowski et al. (2016)*; *Brachycephalus* sp. nov. 1: *Pie et al. (2013)*; *Brachycephalus* sp. 1: *Bornschein et al. (2016a)*; *B. sulfuratus*: *Bornschein et al. (2016b)*, *Ribeiro et al. (2017)*, *Pie et al. (2018b, 2018a)* | Specimen and genetic sequence on GenBank | Specimen examined (MHNCI 10788, MHNCI 11573, MHNCI 11575; Figs. 1A, 1E, and 1I); KX198033.1 analyzed sequence (Fig. 7) |

| Species | Locality and state | Geographical coordinates and altitude | Previous identification | Voucher | Our analysis of the record |
|---|---|---|---|---|---|
| *B. sulfuratus* | Entroncamento Teba, Rio Turvo, municipality of Campina Grande do Sul, Paraná | 25°01′28″S, 48°37′12″W; 785 m a.s.l. | — | Specimens and calls | Specimens (MHNCI 11586–7) and calls examined (MHNCI 219) |
| *B. sulfuratus* | Estância Hidroclimática Recreio da Serra, Serra da Baitaca, municipality of Piraquara, Paraná | 25°27′14″S, 49°00′28″W; 1,150–1,205 m a.s.l. | *B. sulfuratus*: *Bornschein, Pie & Teixeira, 2019* | Specimen | Specimen examined (MHNCI 11591) |
| *B. sulfuratus* | Fazenda Thalia, municipality of Balsa Nova, Paraná | 25°30′58″S, 49°40′12″W; 1,025 m a.s.l. | Without species identification: *Firkowski et al. (2016)*; *Brachycephalus* sp. nov. 1: *Pie et al. (2013)*; *Brachycephalus* sp. 1: *Bornschein et al. (2016a)*; *B. sulfuratus*: *Bornschein et al. (2016b), Ribeiro et al. (2017), Pie et al. (2018b)* | Specimens, calls, and genetic sequence on GenBank | Specimens (MHNCI 11579–81, MHNCI 11582; Figs. 1C, 1D, 1G and 1H) and calls examined (MHNCI 134); KX198032.1 analyzed sequence (Fig. 7) |
| *B. sulfuratus* | near the Jurupará dam, municipality of Piedade, São Paulo | 23°56′30″S, 47°23′45″W; 690 m a.s.l. | *B. sulfuratus*: *Pie et al. (2018b)* | Specimens and calls | Specimens (MHNCI 10790–2; Figs. 1J and 1L) and calls examined (MHNCI 123–5; Figs. 3A, 3C and 3D) |
| *B. sulfuratus* | Mananciais da Serra, municipality of Piraquara, Paraná | 25°29′32″S, 48°59′33″W; 970–1,050 m a.s.l. | *Brachycephalus* sp. nov. 1: *Pie et al. (2013)*; *Brachycephalus* sp. 1: *Bornschein et al. (2016a)*; *B. sulfuratus*: *Bornschein et al. (2016b), Ribeiro et al. (2017), Pie et al. (2018b)* | Specimen | Specimen examined (MHNCI 10302) |
| *B. sulfuratus* | Monte Crista, municipality of Garuva, Santa Catarina | 26°04′53″S; 48°55′03″W; 435 m a.s.l. | — | Calls | Calls examined (MHNCI 221) |
| *B. sulfuratus* | Morro Anhangava, municipality of Quatro Barras, Paraná | 25°22′51″S, 49°01′26″W; 915 m a.s.l. | *B. sulfuratus*: *Condez et al. (2016), Monteiro et al. (2018b)* | Specimen and genetic sequence on GenBank | MK697488.1, MK697440.1, KU321534.1, and MG889428.1 analyzed sequences (Fig. 7) |
| *B. sulfuratus* | Morro do Canal, municipality of Piraquara, Paraná | 25°30′55″S; 48°58′56″W; 1,315 m | — | Calls | Calls examined (MHNCI 220) |
| *B. sulfuratus* | Morro do Cantagalo, Vila da Glória, Distrito do Saí, municipality of São Francisco do Sul, Santa Catarina | 26°10′31″S, 48°42′44″W; 160 m a.s.l. | *B. sulfuratus*: *Condez et al. (2016)* | Specimen and genetic sequence on GenBank | MK697441.1, MK697489.1, KU321532.1, and MK697392.1 analyzed sequences (Fig. 7) |
| *B. sulfuratus* | Morro do Garrafão, municipality of Corupá, Santa Catarina | 26°28′23″S, 49°15′57″W; 500–530 m a.s.l. | *B. sulfuratus*: *Pie et al. (2018b), Teixeira et al. (2018)* | Specimen and calls | Specimens (MHNCI 10826–8; Fig. 1K) and calls examined (MHNCI 137) |

*(Continued)*

| Species | Locality and state | Geographical coordinates and altitude | Previous identification | Voucher | Our analysis of the record |
|---|---|---|---|---|---|
| *B. sulfuratus* | Morro Garuva, municipality of Garuva, Santa Catarina | 26°02′29″S, 48°53′14″W; 215–495 m a.s.l. | *B. sulfuratus*: Bornschein, Pie & Teixeira, 2019 | Calls | Calls examined (MHNCI 136) |
| *B. sulfuratus* | Municipality of Barra do Turvo | c. 24°45′S, 48°29′W; altitude? | *B. sulfuratus*: GenBank | Genetic sequence on GenBank | MK697486.1, MK697438.1, and MK697389.1 analyzed sequences (Fig. 7) |
| *B. sulfuratus* | Municipality of Piedade, São Paulo | c. 23°54′S, 47°25′W; altitude? | *B. hermogenesi*: Condez, Sawaya & Dixo (2009), Clemente-Carvalho et al. (2011); *Brachycephalus* sp. cf. *B. sulfuratus* or *B. hermogenesi*: Bornschein, Pie & Teixeira, 2019 | Specimen and genetic sequence on GenBank | HQ435682.1 and HQ435709.1 analyzed sequences (Fig. 7) |
| *B. sulfuratus* | Núcleo Itutinga-Pilões, Parque Estadual da Serra do Mar, municipality of Cubatão, São Paulo | 23°54′17″S, 46°29′22″W; 55 m a.s.l. | *B. sulfuratus*: Bornschein, Pie & Teixeira, 2019 | Calls | Calls examined (MHNCI 126–7) |
| *B. sulfuratus* | Parque Estadual da Ilha do Cardoso, municipality of Cananéia, São Paulo | 25°06′53″S, 47°55′40″W; 385 m a.s.l. | Possibly *B. hermogenesi*: Verdade et al. (2008); *B. sulfuratus*: Condez et al. (2016) | Specimen, calls, and genetic sequence on GenBank | MK697485.1, MK697437.1, KU321535.1, and MK697388.1 analyzed sequences (Fig. 7) |
| *B. sulfuratus* | Parque Estadual Intervales, municipality of Iporanga, São Paulo | 24°16′33″S, 48°25′04″W; 820 m a.s.l. | *B. sulfuratus*: Bornschein, Pie & Teixeira, 2019 | Calls | Calls examined (XC80463 XC18179, XC75544) |
| *B. sulfuratus* | Pedra da Tartaruga, municipality of Garuva, Santa Catarina | 25°59′42″S, 48°54′23″W; 465 m a.s.l. | — | Specimen | Specimen examined (MHNCI 11585) |
| *B. sulfuratus* | Pico Marumbi, Parque Estadual do Pico Marumbi, municipality of Morretes, Paraná | 25°27′03″S; 48°54′59″W; 1180 m a.s.l. | — | Specimen | Specimen examined (MHNCI 10302) |
| *B. sulfuratus* | Recanto das Hortências, municipality of São José dos Pinhais, Paraná | 25°33′24″S, 48°59′38″W; 975 m a.s.l. | *Brachycephalus* sp. 1: Bornschein et al. (2016a); *B. sulfuratus*: Ribeiro et al. (2017), Bornschein et al. (2016b), Pie et al. (2018b) | Specimen | Specimen examined |
| *B. sulfuratus* | Reserva Particular do Patrimônio Natural Salto Morato, municipality of Guaraqueçaba, Paraná | 25°09′14″S, 48°18′06″W; 40–880 m a.s.l. | *B. hermogenesi*: Pereira et al. (2010), Santos-Pereira et al. (2011, 2016), Santos-Pereira, Pombal & Rocha (2018), Leivas et al. (2018); *Brachycephalus* sp. 1: Bornschein et al. (2016a) | Specimen and calls | Calls examined (MHNCI 133) |
| *B. sulfuratus* | Salto do Inferno, Rio Capivari, municipality of Bocaiúva do Sul, Paraná | 25°00′02″S, 48°37′07″W; 610 m a.s.l. | *B. sulfuratus*: Ribeiro et al. (2017), Bornschein et al. (2016b), Pie et al. (2018b) | Specimen | Specimen examined |

| Species | Locality and state | Geographical coordinates and altitude | Previous identification | Voucher | Our analysis of the record |
|---|---|---|---|---|---|
| *B. sulfuratus* | Serra do Guaraú, on the border of the municipalities of Cajati and Jacupiranga, São Paulo | 24°47′12″S, 48°07′11″W; 680–835 m a.s.l. | *B. sulfuratus*: *Bornschein, Pie & Teixeira, 2019* | Calls | Calls examined (MHNCI 130) |
| *B. sulfuratus* | Serra do Pico, municipality of Joinville, Santa Catarina | 26°08′31″S, 48°57′19″W; 340–720 m a.s.l. | *B. sulfuratus*: *Bornschein, Pie & Teixeira, 2019* | Calls | Calls examined (MHNCI 217) |
| *B. sulfuratus* | Torre Embratel, municipality of Cajati, São Paulo | 24°52′46″S, 48°15′27″W; 960–990 m a.s.l. | *B. sulfuratus*: *Bornschein, Pie & Teixeira, 2019* | Specimen and calls | Specimen (MHNCI 11588) and calls examined (MHNCI 218) |
| *B. sulfuratus* | Truticultura, municipality of Garuva, Paraná | 26°01′33″S, 48°52′02″W; 90 m a.s.l. | *Brachycephalus* sp. nov. 1: *Pie et al. (2013)*; *Brachycephalus* sp. 1: *Bornschein et al. (2016a)*; *B. sulfuratus*: *Bornschein, Pie & Teixeira, 2019* | Calls | Calls examined (MHNCI 135) |
| *B. hermogenesi* | | | | | |
| *B. hermogenesi* | Corcovado, municipality of Ubatuba, São Paulo | 23°28′20″S, 45°11′41″W; 30–250 m a.s.l. | *B. hermogenesi*: *Bornschein, Pie & Teixeira, 2019* ; *in part.*) | Calls | Calls examined (MHNCI 166; Figs. 4A and 4D) |
| *B. hermogenesi* | Estação Biológica de Boracéia, municipality of Salesópolis, São Paulo | 23°39′10″S, 45°53′05″W; 825–900 m a.s.l. | *B. hermogenesi*: *Pimenta, Bérnils & Pombal (2007)*, *Verdade et al. (2008)*, *Pie et al. (2013)*, *Bornschein et al. (2016a)*, *Condez et al. (2016)* | Specimens and calls | Specimens (MHNCI, one uncatalogued specimen) and calls examined (MHNCI 166-9; Fig. 4E), including recordings sent by V. K. Verdade |
| *B. hermogenesi* | Fazenda Capricórnio, municipality of Ubatuba, São Paulo | 23°23′27″S, 45°04′26″W; 60 m a.s.l. | *B. hermogenesi*: *Giaretta & Sawaya (1998)*, *Verdade et al. (2008)*, *Pie et al. (2013)*, *Bornschein et al. (2016a)*, *Condez et al. (2016)* | Specimens (paratypes) | Specimen examined (ZUEC 9725) |
| *B. hermogenesi* | Morro do Cantagalo, municipality of Caraguatatuba, São Paulo | 23°36′23″S, 45°23′34″W; 155-195 m a.s.l. | — | Calls | Calls examined (MHNCI 222-3) |
| *B. hermogenesi* | Municipality of Paraibuna, São Paulo | c. 23°23′34″S, 45°39′42″W; altitude? | *B. hermogenesi*: *Condez et al. (2016)* | Specimen and genetic sequence on GenBank | MK697373.1 analyzed sequence (Fig. 7) |
| *B. hermogenesi* | Núcleo Cunha, Parque Estadual da Serra do Mar, municipality of Cunha, São Paulo | 23°15′48″S, 45°02′39″W; 1,045–1,140 m a.s.l. | *B. hermogenesi*: *Bornschein, Pie & Teixeira, 2019* | Specimen and calls | Specimen (MHNCI, one uncatalogued specimen) and calls examined (MHNCI 170-1) |

(Continued)

| Species | Locality and state | Geographical coordinates and altitude | Previous identification | Voucher | Our analysis of the record |
|---|---|---|---|---|---|
| B. hermogenesi | Núcleo Picinguaba, Parque Estadual da Serra do Mar, municipality of Ubatuba, São Paulo | 23°22′21″S, 44°49′53″W; 0–700 m a.s.l. | B. hermogenesi: Giaretta & Sawaya (1998), Pimenta, Bérnils & Pombal (2007), Verdade et al. (2008), Clemente-Carvalho et al. (2009), Pie et al. (2013), Bornschein et al. (2016a), Condez et al. (2016), Pie et al. (2018b) | Specimens (holotype and paratypes), calls, and genetic sequence on GenBank | Specimens (ZUEC 9715–21; Fig. 3D) and calls examined (MHNCI 172-87; Figs. 4B, 4C and 4F); MK697472.1, KU321531.1, and MK697374.1 analyzed sequences (Fig. 7) |
| B. hermogenesi | Núcleo Santa Virgínea, Parque Estadual da Serra do Mar, municipality of São Luiz do Paraitinga, São Paulo | 23°19′36″S, 45°07′57″W; 915 m a.s.l. | — | Calls | Calls examined (XC253045) |
| B. hermogenesi | Parque Natural Municipal Nascentes de Paranapiacaba, municipality of Santo André, São Paulo | 23°46′10″S, 46°17′36″W; 840 m a.s.l. | B. hermogenesi: Verdade, Rodrigues & Pavan (2009); Brachycephalus sp. cf. B. sulfuratus or B. hermogenesi: Bornschein, Pie & Teixeira, 2019 | Calls | Calls examined (MHNCI 213-6) |
| B. hermogenesi | Sertão da Cutia, municipality of Ubatuba, So Paulo | not located | B. hermogenesi: Condez et al. (2016) | Specimen | — |
| B. hermogenesi | Trilha do Ipiranga 50 m from the Rio Ipiranga, Núcleo Santa Virgínia, Parque Estadual da Serra do Mar, municipality of São Luiz do Paraitinga, São Paulo | 23°20′41″S, 45°08′21″W; 920–940 m a.s.l. | B. hermogenesi: Bornschein, Pie & Teixeira, 2019 | Calls | Calls examined (MHNCI 188-92) |
| *Brachycephalus* sp. (other than *B. sulfuratus* and *B. hermogenesi*) | | | | | |
| Brachycephalus sp. | Corcovado, municipality of Ubatuba, São Paulo | 23°28′20″S, 45°11′41″W; 30–250 m a.s.l. | B. hermogenesi: Giaretta & Sawaya (1998), Verdade et al. (2008), Pie et al. (2013), Bornschein et al. (2016a), Pie et al. (2018b); collected at "Picinguaba" [= Corcovado], Bornschein, Pie & Teixeira, 2019 in part.) | Specimens (including paratypes) and calls | Specimens (ZUEC 9722-4, MHNCI 10823-5) and calls examined (MHNCI 193–205; Figs. 5A–5C) |
| Brachycephalus sp. | Trilha do Corisco, municipality of Paraty, Rio de Janeiro | 23°16′38″S, 44°46′39″W; 350–725 m a.s.l. | B. hermogenesi: Bornschein, Pie & Teixeira, 2019 | Calls | Calls examined (MHNCI 206–12; Fig. 5D) |
| *Brachycephalus* sp. (*B. hermogenesi* or *B. sulfuratus*) | | | | | |
| Brachycephalus sp. | Alto Quiriri, municipality of Garuva, Santa Catarina | 26°05′34″S, 48°59′41″W; 240 m a.s.l. | Brachycephalus sp. nov. 1: Pie et al. (2013); Brachycephalus sp. 1: Bornschein et al. (2016a); Brachycephalus sp. cf. B. sulfuratus: Bornschein, Pie & Teixeira, 2019 | Unvouchered | The calls resemble those of B. sulfuratus (auditory record made by MRB) |

| Species | Locality and state | Geographical coordinates and altitude | Previous identification | Voucher | Our analysis of the record |
|---|---|---|---|---|---|
| *Brachycephalus* sp. | Colônia Castelhanos, municipality of Guaratuba, Paraná | 25°47′58″S, 48°54′40″W; 290 m a.s.l. | *Brachycephalus* aff. *hermogenesi*: *Cunha, Oliveira & Hartmann (2010)*; *B. hermogenesi Oliveira et al. (2011)*; *Brachycephalus* sp. nov. 1: *Pie et al. (2013)*; *Brachycephalus* sp. 1: *Bornschein et al. (2016a)*; *B. sulfuratus*: *Condez et al. (2016)*; *Brachycephalus* sp. cf. *B. sulfuratus*: *Bornschein, Pie & Teixeira, 2019* | Specimen | Specimen examined (ZUEC 16602) |
| *Brachycephalus* sp. | Dona Francisca, municipality of Joinville, Santa Catarina | 26°09′52″S, 48°59′23″W; 150 m a.s.l. | *Brachycephalus* sp. nov. 1: *Pie et al. (2013)*; *Brachycephalus* sp. 1: *Bornschein et al. (2016a)*; *Brachycephalus* sp. cf. *B. sulfuratus*: *Bornschein, Pie & Teixeira, 2019* | Unvouchered | The calls resemble those of *B. sulfuratus* (auditory record made by MRB) |
| *Brachycephalus* sp. | Estação Ecológica Juréia-Itatins, municipality of Iguape, São Paulo | c. 24°27′S, 47°24′W; altitude? | *B. hermogenesi*: *Verdade et al. (2008)*; *Brachycephalus* sp. cf. *B. sulfuratus* or *B. hermogenesi*: *Bornschein, Pie & Teixeira, 2019* | Specimen | — |
| *Brachycephalus* sp. | Estrada do Rio do Júlio, municipality of Joinville, Santa Catarina | 26°17′02″S, 49°06′08″W; 650 m a.s.l. | *Brachycephalus* sp.: *Mariotto (2014)*; *Brachycephalus* sp. 1: *Bornschein et al. (2016a)*; *Brachycephalus* sp. cf. *B. sulfuratus*: *Bornschein, Pie & Teixeira, 2019* | Specimen | — |
| *Brachycephalus* sp. | Fazenda Pico Paraná, municipality of Campina Grande do Sul, Paraná | 25°13′29″S, 48°51′17″W; 1,050–1,085 m a.s.l. | *Brachycephalus* sp. nov. 1: *Pie et al. (2013)*; *Brachycephalus* sp. 1: *Bornschein et al. (2016a)*; *Brachycephalus* sp. cf. *B. sulfuratus*: *Bornschein, Pie & Teixeira, 2019* | Unvouchered | The calls resemble those of *B. sulfuratus* (auditory records made by MRB and LFR) |
| *Brachycephalus* sp. | Fazenda Primavera, municipality of Tunas do Paraná, Paraná | 24°53′08″S, 48°45′51″W; 1,060 m a.s.l. | *Brachycephalus* sp. nov. 1: *Pie et al. (2013)*; *Brachycephalus* sp. 1: *Bornschein et al. (2016a)*; *Brachycephalus* sp. cf. *B. sulfuratus*: *Bornschein, Pie & Teixeira, 2019* | Unvouchered | The calls resemble those of *B. sulfuratus* (auditory record made by MRB) |
| *Brachycephalus* sp. | Municipality of Ibiúna, São Paulo | c. 23°39′S, 47°13′W; altitude? | *B. hermogenesi*: *Condez et al. (2016)*; *Brachycephalus* sp. cf. *B. sulfuratus* or *B. hermogenesi*: *Bornschein, Pie & Teixeira, 2019* | Specimen | — |
| *Brachycephalus* sp. | Municipality of Juquitiba, São Paulo | c. 23°56′S, 47°04′W; altitude? | *B. hermogenesi*: *Verdade et al. (2008)*, *Condez et al. (2016)*; *Brachycephalus* sp. cf. *B. sulfuratus* or *B. hermogenesi*: *Bornschein, Pie & Teixeira, 2019* | Specimen | — |

(Continued)

| Species | Locality and state | Geographical coordinates and altitude | Previous identification | Voucher | Our analysis of the record |
|---|---|---|---|---|---|
| *Brachycephalus* sp. | Municipality of Peruíbe, São Paulo | 24°18′S, 46°59′W; altitude? | B. hermogenesi: *Condez et al. (2016)*; *Brachycephalus* sp. cf. *B. sulfuratus* or *B. hermogenesi*: *Bornschein, Pie & Teixeira, 2019* | Specimen | — |
| *Brachycephalus* sp. | Municipality of Registro, São Paulo | c. 24°30′S, 47°51′W; altitude? | B. hermogenesi: *Condez et al. (2016)*; *Brachycephalus* sp. cf. *B. sulfuratus* or *B. hermogenesi*: *Bornschein, Pie & Teixeira, 2019* | Specimen | — |
| *Brachycephalus* sp. | Municipality of Ribeirão Grande, São Paulo | c. 24°06′S, 48°22′W; altitude? | B. hermogenesi: *Verdade et al. (2008)*; *Brachycephalus* sp. cf. *B. sulfuratus* or *B. hermogenesi*: *Bornschein, Pie & Teixeira, 2019* | Specimen | — |
| *Brachycephalus* sp. | Municipality of Tapiraí, São Paulo | c. 23°57′55″S, 47°30′19″W; 870 m a.s.l. | B. hermogenesi: *Verdade et al. (2008)*, Condez, Sawaya & Dixo (2009); *Brachycephalus* sp. cf. *B. sulfuratus* or *B. hermogenesi*: *Bornschein, Pie & Teixeira, 2019* | Specimen | — |
| *Brachycephalus* sp. | Parque Estadual de Jacupiranga, municipality of Eldorado, São Paulo | c. 24°38′S, 48°24′W; altitude? | B. hermogenesi: *Condez et al. (2016)*; *Brachycephalus* sp. cf. *B. sulfuratus* or *B. hermogenesi*: *Bornschein, Pie & Teixeira, 2019* | Specimen | — |
| *Brachycephalus* sp. | Pico Agudinho, Serra da Prata, municipality of Morretes, Paraná | 25°36′24″S, 48°43′33″W; 385 m a.s.l. | *Brachycephalus* sp. nov. 1: *Pie et al. (2013)*; *Brachycephalus* sp. 1: *Bornschein et al. (2016a)*; *Brachycephalus* sp. cf. *B. sulfuratus*: *Bornschein, Pie & Teixeira, 2019* | Unvouchered | The calls resemble those of *B. sulfuratus* (auditory record made by MRB) |
| *Brachycephalus* sp. | Reserva Betary, municipality of Iporanga, São Paulo | 24°33′08″S, 48°40′49″W; 190 m a.s.l. | *Brachycephalus* sp. cf. *B. sulfuratus* or *B. hermogenesi*: *Bornschein, Pie & Teixeira, 2019* | Specimen | Specimen examined (ZUEC 19931) |
| *Brachycephalus* sp. | Reserva Biológica do Alto da Serra de Paranapiacaba, municipality of Santo André, São Paulo | 23°46′40″S, 46°18′45″W; 800–850 m a.s.l. | B. hermogenesi: *Verdade et al. (2008)*, *Verdade, Rodrigues & Pavan (2009)*; *Brachycephalus* sp. cf. *B. sulfuratus* or *B. hermogenesi*: *Bornschein, Pie & Teixeira, 2019* | Unvouchered | — |
| *Brachycephalus* sp. | Reserva Florestal de Morro Grande, municipality of Cotia, São Paulo | 23°42′08″S, 46°58′22″W; cf. 990 m a.s.l. | B. hermogenesi: *Dixo & Verdade (2006)*, *Verdade et al. (2008)*, *Condez et al. (2016)*; *Brachycephalus* sp. cf. *B. sulfuratus* or *B. hermogenesi*: *Bornschein, Pie & Teixeira, 2019* | Specimen | — |

| Species | Locality and state | Geographical coordinates and altitude | Previous identification | Voucher | Our analysis of the record |
|---|---|---|---|---|---|
| *Brachycephalus* sp. | Sítio Ananias, municipality of Guaratuba, Paraná | 25°47′08″S, 48°43′03″W; 25 m a.s.l. | *Brachycephalus* sp. nov. 1: *Pie et al. (2013)*; *Brachycephalus* sp. 1: *Bornschein et al. (2016a)*; *Brachycephalus* sp. cf. *B. sulfuratus*: *Bornschein, Pie & Teixeira, 2019* | Unvouchered | The calls resemble those of *B. sulfuratus* (auditory record made by MRB) |
| *Brachycephalus* sp. (*B. hermogenesi* or *Brachycephalus* sp. from Corcovado and Trilha do Corisco) | | | | | |
| *Brachycephalus* sp. | Morro Cuscuzeiro, on the border of municipalities of Paraty, Rio de Janeiro, and Ubatuba, São Paulo | 23°17′50″S, 44°47′21″W; 730–1,090 a.s.l. | *B. hermogenesi*: *Bornschein, Pie & Teixeira, 2019* | Unvouchered | The calls resemble those of *Brachycephalus* sp. of Trilha do Corisco (auditory record made by MRB and LFR) |
| *Brachycephalus* sp. | Morro do Corcovado, Parque Estadual da Serra do Mar, municipality of Ubatuba, São Paulo | 23°27′06″S, 45°12′03″W; 250–1,060 m a.s.l. | *B. hermogenesi*: *Bornschein, Pie & Teixeira, 2019* | Unvouchered | The calls resemble those of *Brachycephalus* sp. of Trilha do Corisco (auditory record made by MRB and LFR) |
| *Brachycephalus* sp | Municipality of Paraty, Rio de Janeiro | c. 23°13′07″S, 44°43′15″W; altitude? | *B. hermogenesi*: *Giaretta & Sawaya (1998)*; *Brachycephalus* sp. cf. *B. hermogenesi*: *Bornschein, Pie & Teixeira, 2019* | Unvouchered | — |

**Note:**
Our revision resulted in some unidentified records (*B. sulfuratus*, *B. hermogenesi* or a third species); the probable identifications are provided below. Localities are in alphabetical order (accordingly to the respective species). Abbreviations: FNJV = fonoteca neotropical Jacques Vielliard; MHNCI = Museu de História Natural Capão da Imbuia, Curitiba, Paraná, Brazil; ZUEC = Museu de História Natural, Universidade Estadual de Campinas, Campinas, state of São Paulo, Brazil; XC = Xeno-Canto sound collection (www.xeno-canto.org).

## Diagnosis between *Brachycephalus sulfuratus* and *B. hermogenesi*

*Condez et al. (2016)* indicated three morphological characters to diagnose *B. sulfuratus* from the very similar *B. hermogenesi*: (1) It "differs from… *B. hermogenesi*… by having (in life) yellow blotches on the ventral surfaces of the throat, chest, arms, and forearms" (*Condez et al., 2016*: 43, 50); (2) a more evident "singular inverted v-shaped mark around the cloacal region in ventral view", that is "generally rounded and not ornamented in… *B. hermogenesi*…" (*Condez et al., 2016*: 43, 50); and (3) the presence of an "m-shaped mark around the cloacal opening [in dorsal view], which is… not clearly defined in *B. hermogenesi*" (*Condez et al., 2016*: 50). Specimens of *B. sulfuratus* collected in southern São Paulo, Paraná and Santa Catarina (Table 1) have revealed that the yellow spots on the ventral surface of this species might still be present, on the throat, chest, arms, and/or forearms, but not necessarily in all of these body parts. In addition, the amount of yellow is highly variable, being virtually absent in some individuals (Fig. 1). Moreover, in three individuals of *B. sulfuratus* collected by us in the state of São Paulo (near the Jurupará dam; Table 1), two do not present yellow spots on the ventral surface (see one of them in Fig. 1L), being identified as *B. sulfuratus* by their advertisement calls (MHNCI 123–5;

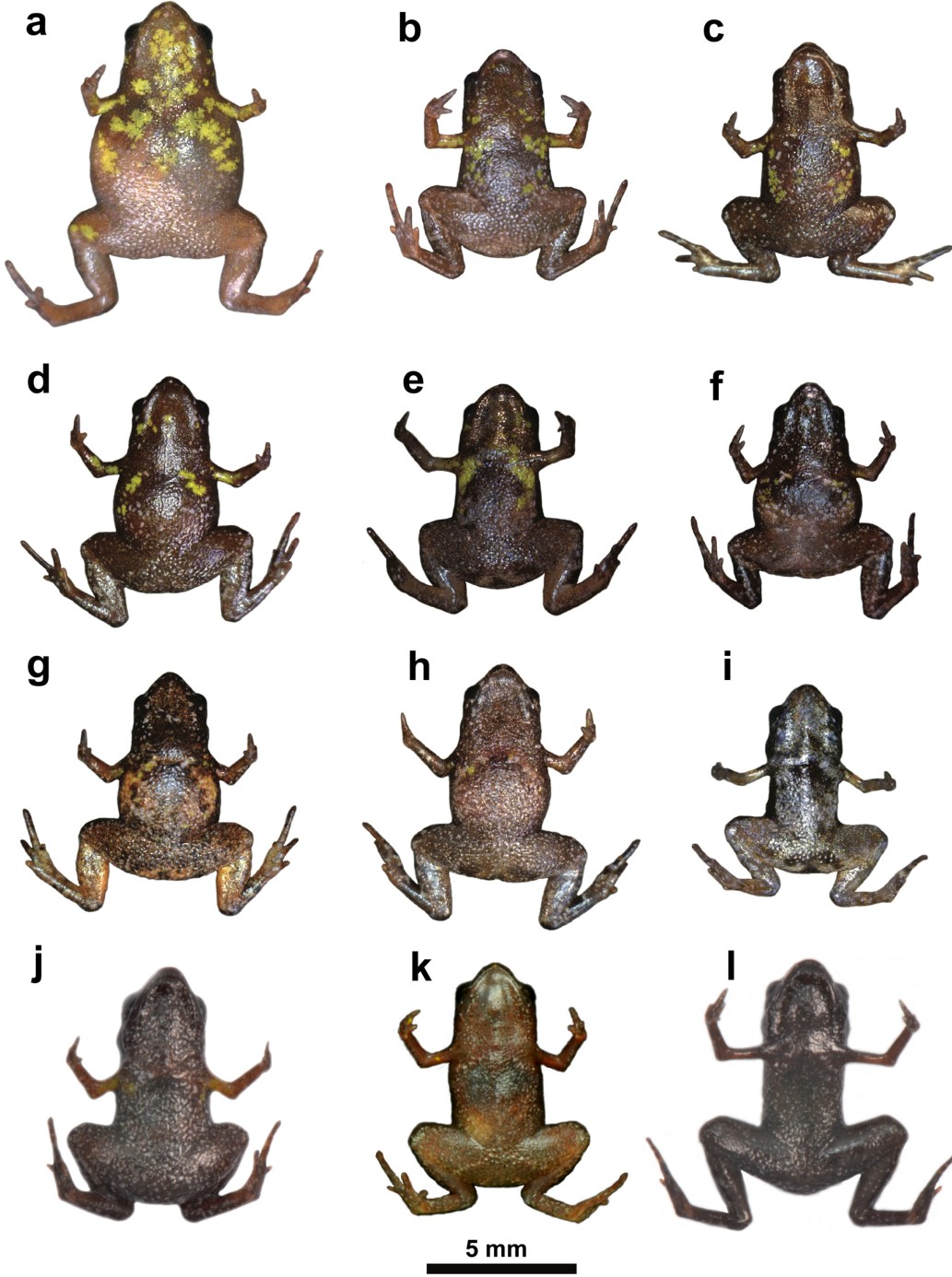

**Figure 1 Ventral view of live specimens of *Brachycephalus sulfuratus*.** Ventral view of live specimens of *Brachycephalus sulfuratus* initially deposited in DZUP) and transferred to MHNCI. (A) MHNCI 11575 (ex-DZUP 153) (Corvo, Paraná); (B) MHNCI 11571 (ex-DZUP 139)(Caratuval, near the Parque Estadual das Lauráceas, Paraná); (C) MHNCI 11582 (ex-DZUP 224) (Fazenda Thalia, Paraná); (D) MHNCI 11579 (ex-DZUP 221) (Fazenda Thalia); (E) MHNCI 11573 (ex-DZUP 151) (Corvo); (F) MHNCI 11583 (ex-DZUP 362) (base of the Serra Água Limpa, São Paulo); (G) MHNCI 11580 (ex-DZUP 222) (Fazenda Thalia); (H) MHNCI 11581 (ex-DZUP 223) (Fazenda Thalia); (I) MHNCI 10788 (ex-DZUP 154) (Corvo); (J) MHNCI 10790 (near the Jurupará dam, São Paulo); (K) MHNCI 10826 (Morro do Garrafão, Santa Catarina); (L) MHNCI 10792 (near the Jurupará dam). Notice the variable of yellow spots, absent
**Figure 1** (continued)
in specimen "l", as well as the absence of the dark-brown inverted v-shaped mark on the cloacal region of
specimen "a". Compare sonograms from specimens "j" and "l" in Figs. 2B and 2C. The presence of yellow
spots and v-shaped mark was proposed as diagnostic characteristics to distinguish *B. sulfuratus* from
*B. hermogenesi*, but they are variable intraspecifically. For details on geographical localities, see Table 1.
Photo credit: Luiz Fernando Ribeiro.               

see below). The inverted v-shaped mark can be absent in individuals of *B. sulfuratus*
(compare Fig. 6A of *Condez et al. (2016)* and Fig. 1A). Additionally, the use of this
character is inconsistent as a diagnosis from *B. hermogenesi* on the actual original
description: "the ventral inverted v-shaped mark… are shared among the four species
(*B. sulfuratus*, *B. hermogenesi*, *B. didactylus* and *B. pulex*)" (*Condez et al., 2016*: 50).
Also, while describing the variation on the type series, the authors stated that "some
individuals present the inverted v-shaped around the cloacal region" (*Condez et al., 2016*:
46). Finally, the "m-shaped mark around the cloacal opening" was also mischaracterized
as a diagnostic character on the actual original description of the species (*Condez et al.,*
*2016*: 50): "The m-shaped mark… are shared among the four species (*B. sulfuratus*,
*B. hermogenesi*, *B. didactylus*, and *B. pulex*)."

   Currently, there are no unique morphological character that could differentiate either
live or preserved specimens (Fig. 2) for *B. sulfuratus* from *B. hermogenesi*. However, for
identification purposes, we considered individuals with yellow spots on their ventral side as
*B. sulfuratus*, whereas individuals without yellow spots could be either *B. sulfuratus* or
*B. hermogenesi*. It is important to note that specimens with yellow spots of *B. sulfuratus*
must be observed in life because in the preservative the change in color prevents separate
them in relation to specimens of *B. hermogenesi*.

   In addition to morphological characters, *Condez et al. (2016*: 43) included in the
diagnosis of *B. sulfuratus* the following parameters of the advertisement call:
"advertisement call long, composed of a set of 4–7 high-frequency notes (6.2–7.2 kHz)
repeated regularly." In the section "Comparisons with other species", *Condez et al. (2016*:
50) stating that "The advertisement call of *B. hermogenesi* is the most similar to the
new species (*B. sulfuratus*), being quite similar in frequency (dominant frequency =
6.8 kHz), which are the highest recorded for the genus. However, the advertisement call of
*B. hermogenesi* can be simple or composed of 2–7 shorter notes with 1–3 pulses (*Verdade*
*et al., 2008*)." In summary, the indicated values overlap with those of *B. hermogenesi*.
The advertisement call of *B. hermogenesi* is composed of 1–7 notes, whereas that of
*B. sulfuratus* is composed of 4–7 notes and the amplitude of the dominant frequency of
*B. hermogenesi* (6.8 kHz) is within the range of *B. sulfuratus* (6.2–7.2).

   These call descriptions do not allow for a reasonable comparison because they are not
necessarily considering the same phenomenon. That is, when it was mentioned that
*B. hermogenesi* call can be simple or composed (*Verdade et al., 2008*), it was being said,
according to the note-centered approach (*Köhler et al., 2017*), that its call can have isolated
notes or note groups, but the total number of notes in the entire *B. hermogenesi* call was
not mentioned. In turn, when mentioning that the *B. sulfuratus* call has 4–7 notes

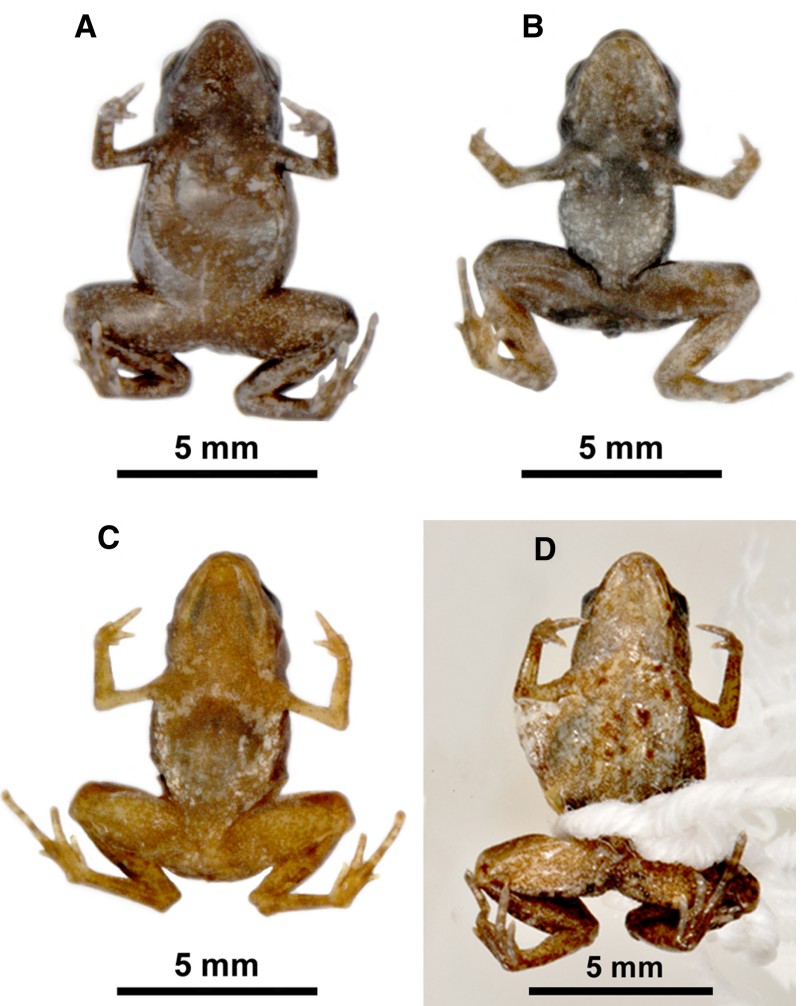

**Figure 2** **Ventral view of specimens of *Brachycephalus sulfuratus* and *B. hermogenesi*.** Ventral view of specimens of *Brachycephalus sulfuratus* (A–C) and *B. hermogenesi* (D) in preservative, deposited in MHNCI and ZUEC: (A) MHNCI 9800 (Salto do Inferno, Paraná); (B) MHNCI 10302 (Mananciais da Serra, Paraná); (C) MHNCI 10303 (Corvo, Paraná; ex DZUP 589); and (D) ZUEC 9715 (Núcleo Picinguaba, São Paulo; holotype of *B. hermogenesi*). Notice the variation in ventral coloration. For details on geographical localities, see Table 1. Photo credit: Luiz Fernando Ribeiro.

(*Condez et al., 2016*), this represents the total number of notes in the call under note-centered approach (sensu *Köhler et al., 2017*) and that all are isolated notes (see *Condez et al., 2016*). This is one notorious distinctions between the calls of *B. sulfuratus* and *B. hermogenesi*: the former presents only isolated notes (Fig. 3) and the latter presents isolated notes and note groups (Fig. 4), with note groups having 2–7 notes, according *Verdade et al. (2008)*, or 2–6 notes, according to our samples (Tables 2 and 3). Other particularities of the call of *B. hermogenesi* in relation to the one of *B. sulfuratus* is the high number of notes per call (≥24) and the presence of "attenuated notes" (Fig. 4F), while in the latter the call has few notes per call (≤8) without attenuated

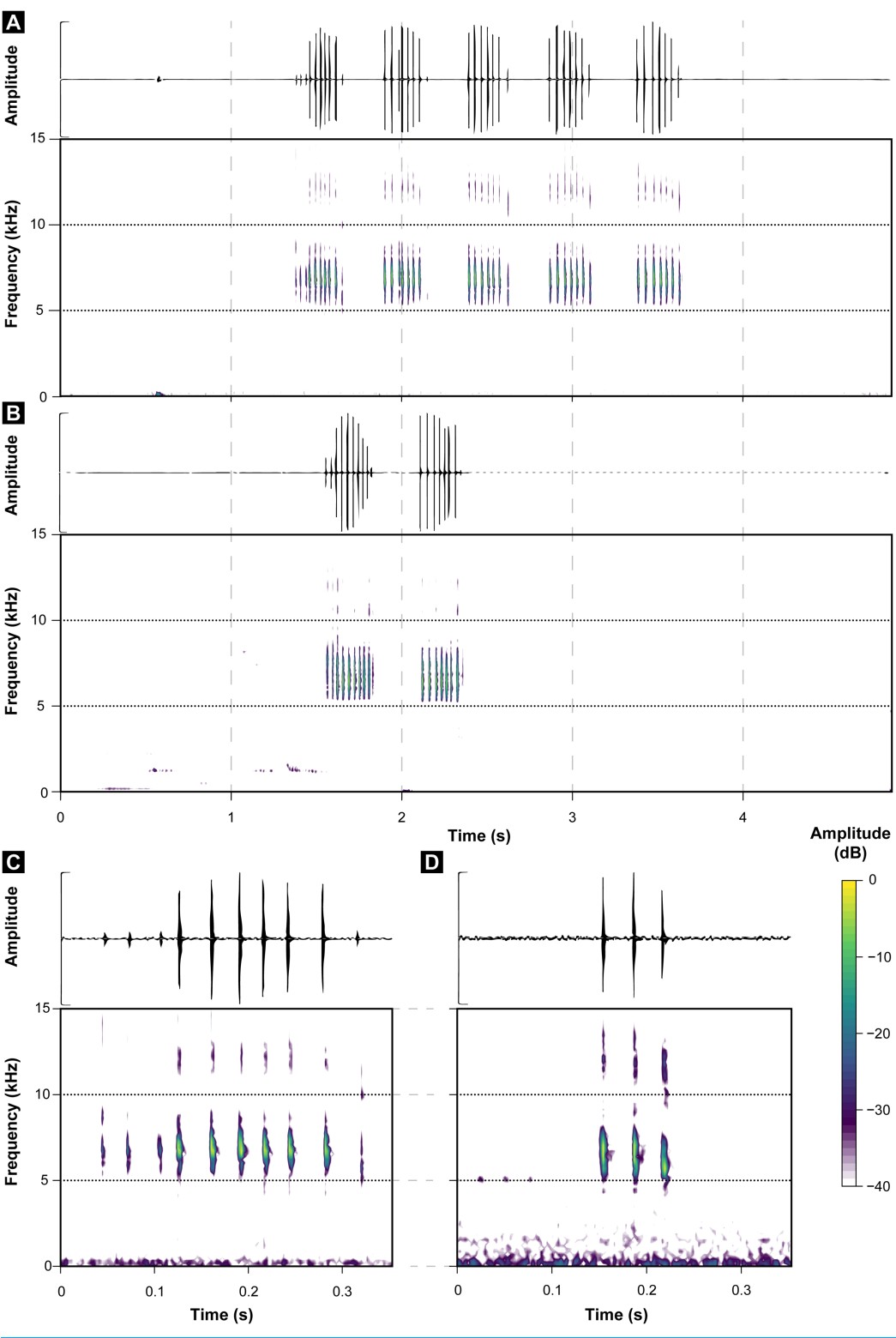

**Figure 3  Oscillograms and spectrograms of *Brachycephalus sulfuratus*.** (A) Example of one entire call with five notes (MHNCI 124; voucher MHNCI 10791 or MHNCI 10792; near the Jurupará dam, municipality of Piedade, São Paulo; M. R. Bornschein). (B) Example of one entire call with two notes (MHNCI 129; voucher MHNCI 11583; Base of the Serra Água Limpa, municipality of Apiaí, São Paulo;

**Figure 3** (continued)
M. R. Bornschein). (C) Example of one note with 10 pulses (MHNCI 124). (D) Example of one note with three pulses (MHNCI 124). Spectrograms are produced with Hann window, overlap of 50%, and FFT size of 512 points in A and B and 256 points in (C) and (D). For details on geographical localities, see Table 1.                                                          

notes (Tables 2 and 3). We introduced attenuated notes as a new parameter, provisionally named, to describe weak notes issued before the notes along the calls of *B. hermogenesi*, more strongly perceived in spectrograms than in oscillograms (Fig. 4F). Due to this attenuated condition and difficulty in perceiving these notes, we did not include them as being part of note groups. We detect the presence of one attenuated note emitted before notes from both isolated notes and note groups, all of which from only three calls (MHNCI 167, MHNCI 183, MHNCI 215; Table 2).

Regarding number of pulses per note, *B. sulfuratus* was described as having 7–11 (*Condez et al., 2016*), but we found 2–14 (Table 1). *Verdade et al. (2008)* have not described the number of pulses of notes of *B. hermogenesi*, as stated by *Condez et al. (2016*: 50; "with 1–3 pulses"*)*. However, as we demonstrated, the number of pulses per note for *B. hermogenesi* is indeed 1–3 (Table 2). We noticed that the calls of individuals of two localities previously attributable of *B. hermogenesi* differs from the descriptions above, by having notes with up to 16 pulses and two or rarely three notes in note groups (Fig. 5; Tables 1–3). These calls were from Trilha do Corisco, municipality of Paraty, Rio de Janeiro state, and Corcovado, municipality of Ubatuba, São Paulo state (see below; Table 1).

We erect as a diagnosis between *B. sulfuratus* and *B. hermogenesi* the few number of notes per call (≤8) with only isolated notes of *B. sulfuratus*, while in *B. hermogenesi* the advertisement call has a high number of notes (≥24) with the presence of isolated notes and note groups (see Table 3). In depth analysis of spectral and temporal parameters of the calls of *B. hermogenesi* will possibly bring other diagnostic parameters, as possibly the note rate, focus of a specific study in the future.

## Geographical occurrence records of *Brachycephalus sulfuratus* and *B. hermogenesi*

Based on our review of the 14 occurrence records of "*Brachycephalus* sp. nov. 1" from *Pie et al. (2013)*, we conclude that the vouchered records correspond to *B. sulfuratus* (Table 1). Specimens from *Pie et al. (2013)* have yellow spots on their ventral side and advertisement calls with few notes and only isolated notes (as above). We treated unvouchered records of *Pie et al. (2013)* as *Brachycephalus* sp. (being probably *B. sulfuratus*; Table 1), with the exception of Castelo dos Bugres, due to the fact that, years later, *Condez et al. (2016)* collected specimens there, confirming the species' identity as *B. sulfuratus*. We also determined previously unidentified *Brachycephalus* records from "Apiaí", "Caratuval", "Corvo" and "Fazenda Thalia" (*Firkowski et al., 2016*) as *B. sulfuratus* (Table 1) based on vouchered identification (specimens had yellow spots on their ventral

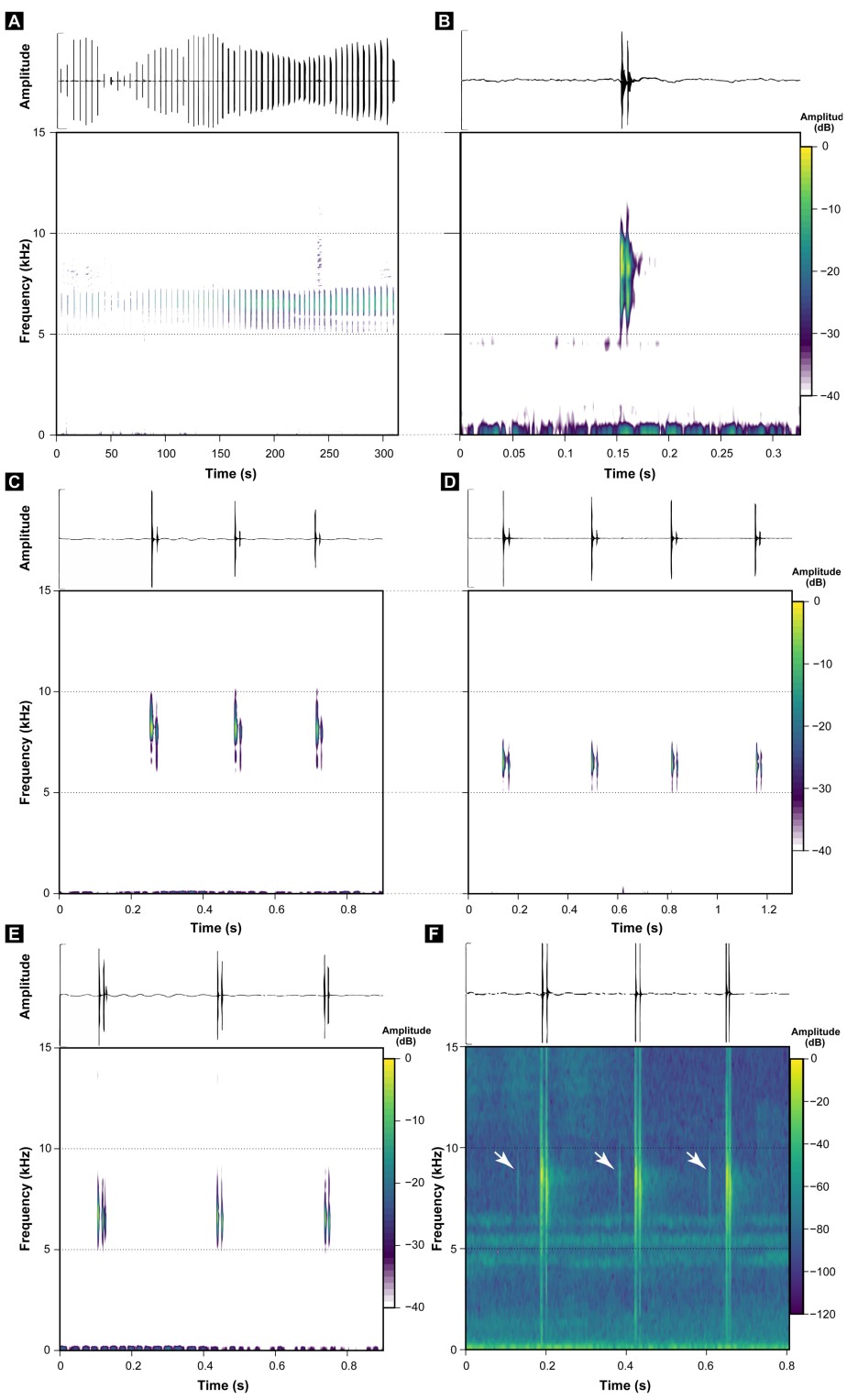

**Figure 4 Oscillograms and spectrograms of *Brachycephalus hermogenesi*.** (A) Example of one entire call with 135 notes recorded (MHNCI 165; Corcovado, municipality of Ubatuba, São Paulo; L. F. Ribeiro). (B) Example of one isolated note with two pulses (MHNCI 183; Núcleo Picinguaba, Parque Estadual da Serra do Mar, municipality of Ubatuba, São Paulo ; M. R. Bornschein). (C) Example of one note group with three notes (each with two pulses; MHNCI 180; Núcleo Picinguaba; M. R. Bornschein).

**Figure 4** (continued)
(D) Example of one note group with four notes (each with two pulses; MHNCI 165). (E) Example of one note group with three notes (the first with three pulses and the remaining with two pulses; MHNCI 166; Estação Biológica de Boracéia, municipality of Salesópolis, São Paulo; M. R. Bornschein). (F) Example of one note group with three notes, with each note preceded by an attenuated note with one pulse (marked with white arrows; MHNCI 183). Spectrograms are produced with Hann window, overlap of 50%, and FFT size of 16,384 points in (A), 128 points in (B) and 256 points in (C)–(F).

region—see Fig. 1). The records of "*Brachycephalus* sp. 1" from *Bornschein et al. (2016a)* correspond to *B. sulfuratus* (Table 1): all but one of them are the same records as those records presented in *Pie et al. (2013)* and *Firkowski et al. (2016)* and were re-identified above. The only exception is the record of "*Brachycephalus* sp. 1" from RPPNSM, municipality of Guaraqueçaba, Paraná, identified as *B. sulfuratus* (Table 1) based on their call structure, with few notes and only isolated notes (MHNCI 133; Table 2). On the basis of this record, we reverted in favor of *B. sulfuratus* all other records of *B. hermogenesi* at RPPNSM (*Pereira et al., 2010*; *Santos-Pereira et al., 2011*, *2016*; *Santos-Pereira, Pombal & Rocha, 2018*; *Leivas et al., 2018*; Table 1).

Some previous studies reporting "*Brachycephalus hermogenesi*" (*Giaretta & Sawaya, 1998*; *Dixo & Verdade, 2006*; *Verdade et al., 2008*; *Condez, Sawaya & Dixo, 2009*; *Verdade, Rodrigues & Pavan, 2009*) from São Paulo do not provide enough morphological evidence or other details to allow us to reassess their original identification by us (Table 1; Fig. 6). Therefore, we propose that these identifications should be reverted as *Brachycephalus* sp. (being *B. hermogenesi* or *B. sulfuratus*). One of these records involves "*B. hermogenesi*" from the municipality of Piedade, state of São Paulo, of *Condez, Sawaya & Dixo (2009)* and *Clemente-Carvalho et al. (2011)*, whose genetic sequence is deposited in GenBank (HQ435682.1 and HQ435709.1; Table 1). The corresponding voucher was obtained by T. H. Condez, 2016, personal communication in her study on the same location (*Condez, Sawaya & Dixo, 2009*). Phylogenetic analyses suggest that it might actually be *B. sulfuratus*, which was placed on the tree together with a specimen from the Municipality of Barra do Turvo, in an early-diverging branch of the *B. sulfuratus* clade on the tree (Fig. 7).

There are some specimens in the original description of *B. sulfuratus* (*Condez et al., 2016*), from six different localities, cited as "*B. hermogenesi*" in the appendix. It is possible that all of these records were identified based on preserved material, which does not allow for proper identification, as indicated above. Therefore, we also propose that those identifications should be considered as *Brachycephalus* sp. (being *B. hermogenesi* or *B. sulfuratus*; Table 1; see also *Bornschein et al., 2016a*).

There is a particular specimen, ZUEC 16602 (see "Introduction"), also examined by us, collected in the state of Paraná, that was first identified as "*Brachycephalus* aff. *hermogenesi*" (*Cunha, Oliveira & Hartmann (2010)*, later as "*B. hermogenesi*" (*Oliveira et al. (2011)*, "*Brachycephalus* sp. nov. 1" (*Pie et al., 2013*), "*Brachycephalus* sp. 1" (*Bornschein et al., 2016a*), and, finally as "*B. sulfuratus*" (*Condez et al., 2016*)). There is also the possibility that this specimen may not have been properly analyzed with respect

**Table 2 Structure of the advertisements calls recording between the geographical distribution of flea toads at some point identified as *B. sulfuratus*, *B. hermogenesi*, and as an unidentified related species.**

| Individuals (Ind) and call deposit number | Call structure | A | B |
|---|---|---|---|
| *B. sulfuratus* | | | |
| Ind 01 (MHNCI 123), ex 01 | 14, 11, 11, 11, 10, 9, 8 | | 0 |
| Ind 01 (MHNCI 123), ex 02 | 12, 10, 11, 10, 10, 9, 8 | | 0 |
| Ind 01 (MHNCI 123), ex 03 | 12, 11, 10, 9, 10, 9, 8 | | 0 |
| Ind 01 (MHNCI 123), ex 04 | 14, 11, 10, 10, 10, 10, 8 | | 0 |
| Ind 02 (MHNCI 124), ex 01 | 10, 7, 6 | | 0 |
| Ind 02 (MHNCI 124), ex 02 | 6, 6, 6, 6 | | 0 |
| Ind 02 (MHNCI 124), ex 03 | 9, 7, 7, 7 | | 0 |
| Ind 02 (MHNCI 124), ex 04 | 10, 7, 8, 7, 3 | | 0 |
| Ind 02 (MHNCI 124), ex 05 | 6, 6, 7, 9, 7, 4 | | 0 |
| Ind 02 (MHNCI 124), ex 06 | 10, 9, 8, 8, 8, 7 | | 0 |
| Ind 02 (MHNCI 124), ex 07 | 10, 9, 8, 9, 9, 8, 7 | | 0 |
| Ind 02 (MHNCI 124), ex 08 | 10, 7, 10, 8, 9, 8 | | 0 |
| Ind 02 (MHNCI 124), ex 09 | 9, 7, 8, 8, 8, 7 | | 0 |
| Ind 02 (MHNCI 124), ex 10 | 10, 8, 7, 7, 8 | | 0 |
| Ind 03 (MHNCI 125), ex 01 | 12, 10, 9, 9, 9, 8 | | 0 |
| Ind 03 (MHNCI 125), ex 02 | 13, 9, 10, 10, 9, 8 | | 0 |
| Ind 03 (MHNCI 125), ex 03 | 10, 9, 9, 9, 9, 9 | | 0 |
| Ind 03 (MHNCI 125), ex 04 | 13, 9, 10, 9, 10, 8 | | 0 |
| Ind 03 (MHNCI 125), ex 05 | 13, 10, 10, 10, 9, 9 | | 0 |
| Ind 03 (MHNCI 125), ex 06 | 11, 9, 10, 10, 9, 8 | | 0 |
| Ind 03 (MHNCI 125), ex 07 | 11, 9, 9, 9, 8 | | 0 |
| Ind 03 (MHNCI 125), ex 08 | 12, 9, 9, 9, 9, 8 | | 0 |
| Ind 04 (MHNCI 126), ex 01 | ?, ?, 9, 8, 8 | | 0 |
| Ind 04 (MHNCI 126), ex 02 | 7, 8, 8, 8, 7 | | 0 |
| Ind 04 (MHNCI 126), ex 03 | 6, 8, 7, 7, 7 | | 0 |
| Ind 04 (MHNCI 126), ex 04 | 6, 8, 8, 8, 8 | | 0 |
| Ind 04 (MHNCI 126), ex 05 | 6, 7, 7, 7, 7 | | 0 |
| Ind 04 (MHNCI 126), ex 06 | 5, 7, 7, 8, 7, 6 | | 0 |
| Ind 05 (MHNCI 127), ex 01 | ?, ?, ?, ? | | 0 |
| Ind 05 (MHNCI 127), ex 02 | ?, ?, ?, ? | | 0 |
| Ind 05 (MHNCI 127), ex 03 | 5, 6, 6, 6, 5 | | 0 |
| Ind 05 (MHNCI 127), ex 04 | ?, ?, ?, ?, ? | | 0 |
| Ind 05 (MHNCI 127), ex 05 | ?, ?, ?, ?, ?, ? | | 0 |
| Ind 05 (MHNCI 127), ex 06 | ?, ?, ?, ?, ? | | 0 |
| Ind 05 (MHNCI 127), ex 07 | 7, 8, 8, 8, 7 | | 0 |
| Ind 06 (MHNCI 128), ex 01 | 11, 10, 10, 9, 8 | | 0 |
| Ind 06 (MHNCI 128), ex 02 | 11, 10, 10, 9, 8 | | 0 |
| Ind 06 (MHNCI 128), ex 03 | 11, 10, 9, 10, 8 | | 0 |
| Ind 06 (MHNCI 128), ex 04 | 12, 10, 9, 9, 8 | | 0 |

| Individuals (Ind) and call deposit number | Call structure | A | B |
|---|---|---|---|
| Ind 06 (MHNCI 128), ex 05 | 11, ?, ?, ? | 0 | |
| Ind 06 (MHNCI 128), ex 06 | 11, 10, 9, 8, 7 | 0 | |
| Ind 06 (MHNCI 128), ex 07 | 11, 10, 9, 9, 9 | 0 | |
| Ind 07 (MHNCI 129), ex 01 | 10, 8 | 0 | |
| Ind 07 (MHNCI 129), ex 02 | 12, 8 | 0 | |
| Ind 07 (MHNCI 129), ex 03 | 10, 8 | 0 | |
| Ind 07 (MHNCI 129), ex 04 | 10, 8, 8 | 0 | |
| Ind 07 (MHNCI 129), ex 05 | 10, 8, 7 | 0 | |
| Ind 08 (MHNCI 129), ex 01 | 6, 5, 4, 4 | 0 | |
| Ind 08 (MHNCI 129), ex 02 | 9, 9, 9, 9 | 0 | |
| Ind 08 (MHNCI 129), ex 03 | 11, 8, 9, 9, 9, 9, 9 | 0 | |
| Ind 08 (MHNCI 129), ex 04 | 9, 9, 7, 7, 9, 9 | 0 | |
| Ind 09 (MHNCI 129) | 10, 9, 9, 9, ?, 9, 8 | 0 | |
| Ind 10 (MHNCI 130), ex 01 | 10, 7, 7, 6 | 0 | |
| Ind 10 (MHNCI 130), ex 02 | 8, 9, 7 | 0 | |
| Ind 11 (MHNCI 130), ex 01 | ?, ?, ?, ?, ?, ? | 0 | |
| Ind 11 (MHNCI 130), ex 02 | ?, ?, ?, ?, ?, ?, ? | 0 | |
| Ind 11 (MHNCI 130), ex 03 | ?, ?, ?, ?, ?, ? | 0 | |
| Ind 11 (MHNCI 130), ex 04 | ?, ?, ?, ?, ? | 0 | |
| Ind 11 (MHNCI 130), ex 05 | 11, 10, 9, 9, 9, 9, 8 | 0 | |
| Ind 11 (MHNCI 130), ex 06 | 12, 9, 9, 9, 9, 9, 8 | 0 | |
| Ind 11 (MHNCI 130), ex 07 | 11, 10, 9, 9, 9, 8 | 0 | |
| Ind 11 (MHNCI 130), ex 08 | 11, 9, 8, 9, 8, 8, | 0 | |
| Ind 11 (MHNCI 130), ex 09 | ?, ?, 9, 9, ?, 8 | 0 | |
| Ind 11 (MHNCI 130), ex 10 | ?, 9, 8, ?, 8, 8 | 0 | |
| Ind 12 (MHNCI 131), ex 01 | 7, 6, 6, 5, 5, 4 | 0 | |
| Ind 12 (MHNCI 131), ex 02 | 7, 6, 5, 6, 7, 5 | 0 | |
| Ind 12 (MHNCI 131), ex 03 | 8, 6, 6, 6, 6, 5 | 0 | |
| Ind 13 (MHNCI 132), ex 01 | 10, 7, 7, 7 | 0 | |
| Ind 13 (MHNCI 132), ex 02 | 9, 8, 8, 8, 8 | 0 | |
| Ind 13 (MHNCI 132), ex 03 | 10, 8, 8, 8, 8 | 0 | |
| Ind 13 (MHNCI 132), ex 04 | 10, 9, 9, 9, 8 | 0 | |
| Ind 13 (MHNCI 132), ex 05 | 10, 9, 9, 9, 9 | 0 | |
| Ind 13 (MHNCI 132), ex 06 | 10, 9, 9, 9, 8 | 0 | |
| Ind 13 (MHNCI 132), ex 07 | 10, 9, 9, 9, 9 | 0 | |
| Ind 13 (MHNCI 132), ex 08 | 11, 9, 9, 9, 9 | 0 | |
| Ind 13 (MHNCI 132), ex 09 | 10, 9, 8, 9, 9 | 0 | |
| Ind 13 (MHNCI 132), ex 10 | 11, 9, 8, 9, 8 | 0 | |
| Ind 13 (MHNCI 132), ex 11 | 10, 9, 10, 8 | 0 | |
| Ind 13 (MHNCI 132), ex 12 | 10, 8, 8, 8 | 0 | |
| Ind 14 (MHNCI 133), ex 01 | ?, ?, ?, ? | 0 | |

| Individuals (Ind) and call deposit number | Call structure | A | B |
|---|---|---|---|
| Ind 14 (MHNCI 133), ex 02 | ?, ?, ?, ? | 0 | |
| Ind 14 (MHNCI 133), ex 03 | ?, ?, ?, ?, ?, ? | 0 | |
| Ind 14 (MHNCI 133), ex 04 | ?, ?, ?, ?, ?, ? | 0 | |
| Ind 14 (MHNCI 133), ex 05 | ?, ?, ?, ?, ?, ? | 0 | |
| Ind 14 (MHNCI 133), ex 06 | 11, 10, 9, 11, 9 | 0 | |
| Ind 14 (MHNCI 133), ex 07 | ?, ?, 10, 9, 8 | 0 | |
| Ind 14 (MHNCI 133), ex 08 | 8, 9, 9, 9, ? | 0 | |
| Ind 14 (MHNCI 133), ex 09 | ?, ?, ?, ? | 0 | |
| Ind 15 (MHNCI 134) | 9, 7, 7, 7, 6, 6 | 0 | |
| Ind 16 (MHNCI 135), ex 01 | 5, 5, 5, 5 | 0 | |
| Ind 16 (MHNCI 135), ex 02 | ?, ?, ?, ?, ? | 0 | |
| Ind 17 (MHNCI 136), ex 01 | 11, 8, 7, 8, 7 | 0 | |
| Ind 17 (MHNCI 136), ex 02 | 12, 9, 8, 8, 8 | 0 | |
| Ind 17 (MHNCI 136), ex 03 | 12, 9, 8, 8, 8 | 0 | |
| Ind 17 (MHNCI 136), ex 04 | 12, 9, 8, 7 | 0 | |
| Ind 17 (MHNCI 136), ex 04 | 10, 9, 8, 5 | 0 | |
| Ind 17 (MHNCI 136), ex 06 | 10, 8, 5, 3 | 0 | |
| Ind 17 (MHNCI 136), ex 07 | 10, 8, 5 | 0 | |
| Ind 17 (MHNCI 136), ex 08 | 9, 8, 6 | 0 | |
| Ind 17 (MHNCI 136), ex 09 | 8, 8, 7 | 0 | |
| Ind 17 (MHNCI 136), ex 10 | 9, 8, 7, 5 | 0 | |
| Ind 18 (MHNCI 137), ex 01 | 6, 7, 6, 2 | 0 | |
| Ind 18 (MHNCI 137), ex 02 | 6, 7, 6, 2 | 0 | |
| Ind 18 (MHNCI 137), ex 03 | ?, 7, 7, 6 | 0 | |
| Ind 18 (MHNCI 137), ex 04 | 8, 7, 8, 7 | 0 | |
| Ind 19 (MHNCI 217), ex. 01 | ?, ?, 10, 10, 9 | 0 | |
| Ind 19 (MHNCI 217), ex. 02 | 9, 10, 10, 9, 10 | 0 | |
| Ind 20 (MHNCI 218), ex 01 | ?, 10, 10, ?, ?, ? | 0 | |
| Ind 20 (MHNCI 218), ex 02 | ?, ?, ?, ?, ?, ? | 0 | |
| Ind 21 (MHNCI 219), ex 01 | 9, 7, 7 | 0 | |
| Ind 21 (MHNCI 219), ex 02 | 9, 7, 7, 6 | 0 | |
| Ind 21 (MHNCI 219), ex 03 | 9, 7, 7, 7 | 0 | |
| Ind 21 (MHNCI 219), ex 04 | 9, 9, 8, 8, 8 | 0 | |
| Ind 21 (MHNCI 219), ex 05 | 10, 8, 8, 8, 8, 8, 8 | 0 | |
| Ind 21 (MHNCI 219), ex 06 | 10, 9, 9, 8, 8, 8 | 0 | |
| Ind 21 (MHNCI 219), ex 07 | 10, 9, 9, 8, 8, 8 | 0 | |
| Ind 21 (MHNCI 219), ex 08 | 10, 9, 9, 9, 8 | 0 | |
| Ind 21 (MHNCI 219), ex 09 | 10, 9, 9, 9, 9, 9, 9, 8 | 0 | |
| Ind 21 (MHNCI 219), ex 10 | 9, 9, 8, 8 | 0 | |
| Ind 21 (MHNCI 219), ex 11 | 10, 8, 7 | 0 | |
| Ind 21 (MHNCI 219), ex 12 | 10, 8, 6 | 0 | |

(Continued)

| Individuals (Ind) and call deposit number | Call structure | A | B |
|---|---|---|---|
| Ind 21 (MHNCI 219), ex 13 | 9, 7, 6 | 0 | |
| Ind 21 (MHNCI 219), ex 14 | 9, 8, 7 | 0 | |
| Ind 21 (MHNCI 219), ex 15 | 10, 8, 7 | 0 | |
| Ind 21 (MHNCI 219), ex 16 | 10, 8, 7 | 0 | |
| Ind 21 (MHNCI 219), ex 17 | 10, 8, 7 | 0 | |
| Ind 21 (MHNCI 219), ex 18 | 10, 8, 7 | 0 | |
| Ind 21 (MHNCI 219), ex 19 | 10, 9, 8 | 0 | |
| Ind 21 (MHNCI 219), ex 20 | 10, 9, 8 | 0 | |
| Ind 21 (MHNCI 219), ex 21 | 10, 9, 8, 8 | 0 | |
| Ind 21 (MHNCI 219), ex 22 | 10, 9, 9, 8 | 0 | |
| Ind 21 (MHNCI 219), ex 23 | 10, 9, 8 | 0 | |
| Ind 21 (MHNCI 219), ex 24 | 10, 9, 8 | 0 | |
| Ind 21 (MHNCI 219), ex 25 | 10, 9, 8 | 0 | |
| Ind 22 (MHNCI 220), ex 01 | 11, 8, 7, 7, 7, 7 | 0 | |
| Ind 22 (MHNCI 220), ex 02 | 10, 8, 7, 7, 8, 8 | 0 | |
| Ind 22 (MHNCI 220), ex 03 | 9, 8, 7, 7, 8, 7 | 0 | |
| Ind 22 (MHNCI 220), ex 04 | 9, 8, 7, 8, 7, 7 | 0 | |
| Ind 22 (MHNCI 220), ex 05 | 10, 8, 8, 8, 8, 8 | 0 | |
| Ind 22 (MHNCI 220), ex 06 | 9, 8, 8, 8, 8, 8 | 0 | |
| Ind 22 (MHNCI 220), ex 07 | 10, 8, 8, 8, 8, 8 | 0 | |
| Ind 22 (MHNCI 220), ex 08 | 10, 8, 8, 8, 8, 8 | 0 | |
| Ind 22 (MHNCI 220), ex 09 | 10, 8, 8, 8, 8 | 0 | |
| Ind 22 (MHNCI 220), ex 10 | 10, 8, 9, 8, 8, 8 | 0 | |
| Ind 22 (MHNCI 220), ex 11 | 10, 8, 7, 8, 8, 7 | 0 | |
| Ind 22 (MHNCI 220), ex 12 | 10, 8, 8, 8, 8, 8 | 0 | |
| Ind 22 (MHNCI 220), ex 13 | 10, 8, 8, 8, 7, 7 | 0 | |
| Ind 22 (MHNCI 220), ex 14 | 10, 8, 8, 8, 8, 6 | 0 | |
| Ind 22 (MHNCI 220), ex 15 | 10, 8, 8, 8, 8, 7 | 0 | |
| Ind 22 (MHNCI 220), ex 16 | 9, 8, 7, 8, 7 | 0 | |
| Ind 22 (MHNCI 220), ex 17 | 10, 9, 7, 8, 7 | 0 | |
| Ind 23 (MHNCI 221), ex 01 | 8, 7, 6, 7, 6, 5 | 0 | |
| Ind 23 (MHNCI 221), ex 02 | 8, 7, 7, 7, 7, 4 | 0 | |
| Ind 23 (MHNCI 221), ex 03 | 8, 7, 7, 7, 6 | 0 | |
| Ind 23 (MHNCI 221), ex 04 | 8, 6, 7, 7, 6 | 0 | |
| Ind 23 (MHNCI 221), ex 05 | 8, 7, 7, 7, 6 | 0 | |
| Ind 23 (MHNCI 221), ex 06 | 8, 7, 8, 7, 7 | 0 | |
| Ind 23 (MHNCI 221), ex 07 | 8, 8, 7, 7, 7 | 0 | |
| Ind 23 (MHNCI 221), ex 08 | 8, 8, 8, 7, 7 | 0 | |
| Ind 23 (MHNCI 221), ex 09 | 9, 8, 7, 8, 7 | 0 | |
| Ind 23 (MHNCI 221), ex 10 | 9, 8, 8, 8, 7 | 0 | |
| Ind 23 (MHNCI 221), ex 11 | 9, 8, 7, 7 | 0 | |

| Table 2 (continued) | | | |
|---|---|---|---|
| **Individuals (Ind) and call deposit number** | **Call structure** | **A** | **B** |
| Ind 23 (MHNCI 221), ex 12 | 8, 7, 6 | 0 | |
| Ind 23 (MHNCI 221), ex 13 | 9, 7, 7, 6 | 0 | |
| Ind 23 (MHNCI 221), ex 14 | 8, 7, 7, 7, 7 | 0 | |
| Ind 23 (MHNCI 221), ex 15 | 9, 7, 7, 7, 7 | 0 | |
| Ind 23 (MHNCI 221), ex 16 | 8, 7, 8, 7, 6 | 0 | |
| Ind 23 (MHNCI 221), ex 17 | 8, 8, 8, 7, 7 | 0 | |
| Ind 23 (MHNCI 221), ex 18 | 9, 8, 7, 8 | 0 | |
| Ind 23 (MHNCI 221), ex 19 | 9, 8, 8, 7, 7 | 0 | |
| Ind 23 (MHNCI 221), ex 20 | 9, 8, 7, 7, 7 | 0 | |
| Ind 23 (MHNCI 221), ex 21 | 8, 7, 7, 7, 7 | 0 | |
| Ind 23 (MHNCI 221), ex 22 | 9, 8, 7, 7 | 0 | |
| Ind 23 (MHNCI 221), ex 23 | 9, 7, 7, 6 | 0 | |
| Ind 23 (MHNCI 221), ex 24 | 9, 7, 7, 7 | 0 | |
| *B. hermogenesi* (Corcovado) | | | |
| Ind 01 (MHNCI 165) | 2, 2, 2, 2, 2, 2, 2, 2, 2, 2, 2, 2, 2, 2, 2, 2, 2, 2, 2, 2, 2, 2, 2, 2, 2, 2, 2, (2-2), (2-2), (2-2), (2-2), (2-2-2), (2-2-2), (2-2-2), (2-2-2-2), (2-2-2), (2-2-2-2), (2-2-2), (2-2-2-2), (2-2-2-2), (2-2-2-2), (2-2-2-2), (2-2-2), (2-2-2), (2-2-2-2), (2-2-2), (2-2-2-2), (2-2-2-2), (2-2-2-2), (2-2-2), (2-2-2-2), (2-2-2), (2-2-2-2), (2-2-2-2), (2-2-2), (2-2-2-2), (2-2-2), (2-2-2), (2-2-2) | 2 | |
| *B. hermogenesi* (Estação Biológica de Boracéia) | | | |
| Ind 01 (MHNCI 166) | 2, 2, 2, 2, 2, 2, 2, 2, 2, 2, 2, 2, 2, 2, (2-2), 2, 2, 2, (2-2), (2-2), (2-2), (2-2), (2-2), (2-2), (2-2), (2-2), (2-2-2), (2-2-2), (2-2-2), (2-2-2), (2-2-2-1), (2-2-2), (2-2-2), (2-2-2), (2-2-2-2), (2-2-2-2), (2-2-2-2), (2-2-1-1), (2-2-2-2), (2-2-2-2), (2-2-2-2), (3-2-2-2), (2-2-2-2), (2-2-2), (2-2-2-2), (3-2-2-2), (2-2-2), (2-2-2), (2-2-2), (2-2-2), (3-2-2), (2-2-2) | 6 | |
| Ind 02 (MHNCI 167) | 2, 1, 2, 2, 1, 2, 2, 2, 1, 2, 2, 2, 2, 2, 2, 2, 2, 2, 2, 2, 2, (2-2), (2-2), (2-2), (2-2), (2-2), (2-2-2), (2-2-2), (2-2-2), (2-2-2), (2-2-1), (2-2-2), (2-2-1), (2-1-2-2), (2-2-1), (2-2-1-1), (2-2-2), (2-2-2), (2-2-2), (2-2-1-1), (2-1-1-₁-2), (2-1-₁-1), (2-1-₁-1-₁-1), (2-2-1) | 7 | |
| Ind 03 (MHNCI 168) | 2, 2, 2, 2, 2, 2, 2, 2, 2, 2, 2, 2, 2, 2, 2, 2, 2, 2, 2, (2-2), (2-2), (2-2), (2-2-2), (2-2-2), (2-2-2), (2-2-2), (2-2-2-2), (2-2-2-2-2), (2-2-2-2), (2-2-2), (2-2-2-2), (2-2-2-2), (2-2-2), (2-2-2-2), (2-2-2-2) | 3 | |
| Ind 04 (MHNCI 169) | 1, 1, 1, 1, 1, 1, 1, 1, 1, 1, 1, 2, 2, ?, ?, 2, 2, 2, 2, (2-2), (2-2-2), (2-2), (2-2), (2-2), (2-2), (2-2), (2-?-?), (2-2-2), (2-2), (2-2-2), (2-2-2), (2-?-2), (2-?-2), (2-2-2-1), (2-2-1), (2-2-2-2), (?-?-?-?), (2-?-?-?), (2-1-1), (2-2-2), (2-2-2), (?-2-?), (2-1-2), (2-1-2) | 3 | |
| *B. hermogenesi* (Morro do Cantagalo) | | | |
| Ind 01 (MHNCI 222) | ?, ?, ?, ?, 2, 2, ?, 2, 2, 2 | ? | X |
| Ind 02 (MHNCI 223) | 2, (2-2-2), (2-2-2), (2-2-2), (2-2-2), (2-2-2-2), (2-2-2-2), (2-2-2-2), (2-2-2-2) | ? | X |
| *B. hermogenesi* (Núcleo Cunha) | | | |
| Ind 01 (MHNCI 170), ex 01 | 2, 2, 2, 2, 2, 2, 2, 2, 2, 2, 2, 2, 2, (2-2), 2, 2, (2-2), (2-2), (2-2), (2-2) | 0 | |
| Ind 01 (MHNCI 171), ex 02 | 1, 1, 1, 1, 2, 2, 2, 2, 2, 2, 2, 2, 2, 2, ?, 2, 2, 2, 2, 2, 2, 2, 2, (2-2), 2, (2-2), (2-?), 2, (2-2), (2-2), (2-2), (2-2) | 0 | |
| *B. hermogenesi* (Núcleo Picinguaba) | | | |
| Ind 01 (MHNCI 172), ex 01 | 2, 2, 2, 2, 2, 2, 2, 2, 2, 2, 2, 2, 2, 2, 2, 2, (2-2), (2-2), (2-2), (2-2), (2-2), (2-2), (2-2), (2-2), (2-2), (2-2-2), (2-2-2), (2-2-2), (2-2-2), (2-2-2), (2-2-2), (2-2-2), (2-2-1), (2-2), (2-2), (2-2), (2-2) | ? | |
| Ind 01(MHNCI 175), ex 02 | 1, 2, 1, 1, 2, 2, 2, 2, 2, 2, 2, 2, 2, 2, 2, 2, 2, 2, 2, 2, (2-1), (2-2), (2-2), (2-2), (2-2), (2-2), (2-2), (2-2) | 0 | |
| Ind 02 (MHNCI 173), ex 01 | 2, 2, 2, 2, 2, 2, 2, (2-1), (2-2), (2-2), (2-2), (2-2), (2-2), (2-2), (2-2-1), (2-1-1), (2-2-2), (2-1-2), (2-2), (2-2-1), (2-2-1), (2-1), (2-1), (1-1), (1-1), (1-1), 1 | 4 | |

(Continued)

| Table 2 (continued) | | | |
|---|---|---|---|
| **Individuals (Ind) and call deposit number** | **Call structure** | **A** | **B** |
| Ind 02 (MHNCI 177), ex 02 | 1, 2, 2, 2, 2, 2, 2, 2, 2, 2, 2, 2, 2, 2, (2-2), 2, (2-2), (2-2), (2-1), (2-2), (2-2), (2-2-2), (2-2-2), (2-2-2), (2-2-2), (2-2-1), (2-1), (2-2), (1-1), (1-1) | 0 | |
| Ind 02 (MHNCI 182), ex 03 | (2-2), (2-2), (2-2), (2-2), (2-2), (2-2) | | ? |
| Ind 03 (MHNCI 174), ex 01 | 2, 2, 2, 2, 2, 2, 2, 2, 2, 2, 2, 2, 2, 2, 2, 2, (2-2), (2-2), (2-2), (2-2), (2-2), (2-2), (2-2), (2-2), (2-2-2), (2-2-2), (2-2-2), (2-2), (2-2-2), (2-2-2), (2-2-2), (2-2-2), (2-2-2), (2-2), (2-2-2), (2-2-1), (2-2) | 6 | |
| Ind 03 (MHNCI 178), ex 02 | 2, 2, 2, 2, 2, 2, 2, 2, 2, 2, 2, 2, (2-2), (2-2), (2-2), (2-2), (2-2), (2-2), (2-2), (2-2), (2-2), (2-2-2), (2-2-2), (2-2-2), (2-2-2), (2-2-2), (2-2-2), (2-2), (2-2-2) | 5 | |
| Ind 03 (MHNCI 180), ex 03 | 2, 2, 2, 2, 2, 2, 2, 2, (2-2), (2-2), (2-2), (2-2), (2-2), (2-2-2), (2-2), (2-2-2), (2-2-2), (2-2-2), (2-2-2), (2-2-2), (2-2-2), (2-2-2), (2-2-2), (2-2-2), (2-2-1), (2-2-2), (2-2-1), (2-2-1), (2-2-1), (2-2-1) | ? | |
| Ind 03 (MHNCI 181), ex 04 | ?, 2, 2, 2, 2, 2, 2, 2, 2, 2, 2, 2, 2, 2, 2, 2, 2, 2, (2-2), (2-2), (2-2), (2-2), (2-2), (2-2), (2-2), (2-2), (2-2), (2-2), (2-2-2), (2-2), (2-2), (2-2-2), (2-2-2), (2-2-2), (2-2-2), (2-2-2), (2-2-1) | 1 | |
| Ind 04 (MHNCI 176), ex 01 | 2, 2, 2, 2, 2, 2, 2, 2, 2, 2, 2, (2-2-1), (2-2-2), (2-2-1), (2-2-2), (2-2-1), (2-2-2), (2-2-1), (2-2-1), (2-2-1), (2-2-1), (2-2-1), (2-2), (2-2-1), (2-2-1), (2-2-1), (2-2), (2-2), (2-2) | 3 | |
| Ind 04 (MHNCI 179), ex 02 | (2-2), (2-2-2), (2-2-1), (2-2-1), (2-2-1), (2-2-2), (2-2-1), (2-2), (2-1) | | ? |
| Ind 04 (MHNCI 183), ex 03 | 1, 1, 1, 1, 1, 2, 2, 2, 2, 2, 2, 2, (2-1), (2-1), (2-1), (2-2), (2-2), (2-2), (2-2), (2-2), (2-2), (2-2-2), (2-2), (2-2), (2-2-2), (2-2-2), (2-2-1), (2-2-1), (2-₁-2-₁2), (2-₁-2-₁2), (2-₁-2-₁2), (₁2-₁-2-₁2), (₁2-₁-2-₁2), (₁2-₁-2-₁1), (2-₁2-1), (2-1) | 0 | |
| Ind 05 (MHNCI 184) | (2-2-2), (2-2-2), (2-2-2), (2-2-2), (2-2-2), (2-2-2), (2-2-1), (2-2-2), (2-2-1) | | ? |
| Ind 06 (MHNCI 185) | 1, 1, 2, 2, 1, 1, 2, 2, 2, 2, 2, (2-1), (2-1), 2, (2-1), 2, (2-2), (2-2), (2-2), (2-2), (2-2), (2-1), (2-1), (2-1), (2-2), (2-2), (2-2) | 2 | |
| Ind 07 (MHNCI 186) | 1, 2, 2, 2, 2, 2, 2, 2, 2, 1, 2, 2, 2, 2, 1, 2, 2, 2, 2, 2, (2-2), (2-2), (2-2), (2-2), (2-2), (2-2), (2-2), (2-2), (2-2), (2-2), (2-2), (2-2), (2-2), 2, 2 | 1 | |
| Ind 08 (MHNCI 187) | (2-2-1), (2-2), (2-2), (2-2), (?-?-?), (2-2), (2-2), (2-2), (2-1), (2-2) | | ? |
| *B. hermogenesi* (Parque Natural Municipal Nascentes de Paranapiacaba) | | | |
| Ind 01 (MHNCI 213) | (?-?-?), (?-?-?), (2-2-2), (2-2-2), (2-2-2), (2-2-2), (2-2-2), (2-2-2), (?-?-?), (?-?-?-?), (?-?-?-?), (?-?-?-?), (?-?-?-?), (1-1-1-1) | | ? |
| Ind 02 (MHNCI 214) | 1, 1, 1, 1, 1, 1, 1, 1, 1, (1-1), 1, 1, 1, 1, 1, 1, 1, 1, 1, 1, 1, 1, 1, 1, 1, 1, 1, ,1, 2, 1, 1, 1, 2, 2, 1, 2, 2, 2, 1, 2, 2, 2, 1, 1, 1, 1, 1, 1, 2, 1, 1, 2, 1, 2, 2, 2, 2, 2, 2, 2, 2, 2, 2, 2, 2, 2, 2, ,2, 2, 2, 2, 2, 2, 2, 2, 2, 2, 2, 2, 2, (2-2), 2, 2, 2, 2, (2-2), 2, 2 | 0 | |
| Ind 03 (MHNCI 215) | 1, 1, 1, 1, 1, 1, 1, 1, 1, 1, 1, 2, 2, 1, 3, 2, 2, 2, 3, 2, 2, 2, 1, 2, 1, 1, 2, 2, 2, 2, 2, ₁2, (2-2), (2-2), (2-2), (2-2), (2-2), (1-2-2), (2-2-2), (2-2-2), (1-2-2), (2-2-2), (2-2-2), (1-1-2), (2-2-2), (2-2-2), (2-2-2), (2-2-1-1), (2-1-1-1), (2-1-1-2), (2-2-1-1), (2-2-1-2), (2-2-1), (1-2-1-2), (2-2-2-1) | 0 | |
| Ind 04 (MHNCI 216) | 1, 1, 1, 1, 1, 1, 1, 1, 1, 1, 1, 1, 1, 1, 1, 1, 1, 1, 1, 1, 1, 1, 1, 2, 1, 1, 1, 1, 1, 1, 1, 2, 1, 2, 1, 1, 2, 2, 1, 2, 2, 1, 2, 2, 2, 2, 2, 2, (2-2), (2-2), 2, (2-2), (2-2), (2-2), (2-2), (2-2), (2-2), (2-2), (2-2-2), (2-2), (2-1-1), (1-2-1), (2-2-1), (2-1-1), (2-1-1), (2-2-1), (1-1-1-1), (2-1-1-1), (1-1) | 0 | |
| *B. hermogenesi* (Trilha do Ipiranga 50 m from the Rio Ipiranga) | | | |
| Ind 01 (MHNCI 188) | (2-2), (2-2), (2-2), (2-1-1), (2-1-1), (1-1-1-1), (1-1-1), (1-1-1-1), (1-1-1-1), (1-1-1), (1-1-1), (1-1-1), (1-1-1) | | ? |
| Ind 02 (MHNCI 189) | 2, 2, 2, 2, 2, 2, 2, 2, 2, 2, 2, 2, 2, 2, 2, 2, 2, ?, ?, 2, 2, (1-2), (2-2), (2-2), (2-2), (3-2), (2-2-2), (2-2), (2-2-2), (2-2-2), (2-2-2), (2-2-2), (2-2-2), (2-2-1), (2-2-2), (2-2), (2-2-2) | 3 | |
| Ind 03 (MHNCI 190) | 2, 2, 2, 2, 2, 2, 2, 2, 2, 2, 2, 2, 2, (2-2), (2-2), (2-2), (2-2), (2-2), (2-2), (2-2), (2-2), (2-2), (2-2-2) | | 3 |
| Ind 04 (MHNCI 191) | 2, 2, 2, 2, 2, 2, 2, 2, 2, 2, 2, 2, 2, 2, 2, 2, (2-2), (2-2), (2-2), (2-2), (2-2), (2-2), (2-2), (2-2), (2-2), (2-2), (2-2) | | 4 |

| Table 2 (continued) | | | |
|---|---|---|---|
| **Individuals (Ind) and call deposit number** | **Call structure** | **A** | **B** |
| Ind 05 (MHNCI 192) | 2, 2, 2, 2, 2, 2, 2, 2, 2, 2, 2, 2, 2, 2, 2, 2, 2, 2, 2, 2, 2, 2, 2, 2, 2, 2, 2, 2, 2, 2, 2, 2, 2, 2, 2, 2, 2, 2, 2, 2, 2, 2, 2, 2, 2, 2, 2, 2, 2, 2, 2, 2, (2-2), (2-2), (2-2), (2-2), (2-2), (2-2), (2-2), (2-2), (2-2-2), (2-2), (2-2-2), (2-2-2), (2-2-2), (2-2-2), (2-2-2-2), (2-2-2-2), (2-2-2-1), (2-2-1-2), (2-2-2-1), (2-1-1-1-1), (2-1-1-1-1), (2-1-1-1-1), (2-2-2-1-1), (1-1-1-1-1), (1-1-1-1-1-1), (2-1-1-1-1), (2-2-1-1-1), (1-2-1-1-1), (1-1-1-1-1), (1-1-1-1-1), (1-1-1-1-1), (1-1-1-1-1), (1-1-1-1-1), (1-2-1-1-1), (1-1-1-1-1), (1-2-1-1-1), (1-1-1-1-1), (1-1-1-1-1), (1-1-1-1), (1-1-1-1), (1-1-1-1-1), (2-1-1-1-1), (2-1-2-1-1), (1-2-1-2-1), (2-1-1-2-1), (1-1-1-1-1), (1-2-1-1-2), (1-1-1-1-1), (1-1-1-1), (1-1-1-1-1) | 3 | |
| *Brachycephalus* sp. (Corcovado) | | | |
| Ind 01 (MHNCI 193) | (6-4), (6-4), (6-4), (6-4), (6-4), (6-4), (6-1) | ? | |
| Ind 02 (MHNCI 194) | 1, 2, 2, ?, 2 | 0 | X |
| Ind 03 (MHNCI 195) | 3, 3, 4, 3, 4, 4, 3, 3, 4, 4, 3, 4, 4, 4, 4, 4, 4, 4, 4, 4, 4, 5, 5, 4, 5, 4, 4, (5-2), (5-4), (5-2), (5-3), (5-2), 6, 5, 4, (5-1), 5, (5-3), (5-3), (5-1), (6-4), (5-3), (5-4), (5-3), (5-3), (5-3), (5-3), (5-3), (5-4), (5-3), 6 | ? | |
| Ind 04 (MHNCI 196), ex 01 | 3, 3, 2, 3, ?, 3, ?, ?, ?, ?, ?, 4, 4, 4, 4, 3, 3, ?, 4, ?, 3, 4, 4, 5, 4, 4, 5, 4, 4, ?, ?, 5, 5, 5, 5, (4-3), (?-?), (6-4), (7-4), (9-4), (8-4), (9-5), (9-4), (10-5), (9-6), (11-5), (11-5), (8-5), (6-5), (6-4), (7-4), (6-4), 6, 5, 5, 5, 5, 5, 5, 5 | 3 | |
| Ind 04 (MHNCI 200), ex 02 | 4, 4, 4, 5, 5, 4, 5, 4, 4, 4, 4, 5, 5, 5, (4-3), 5, 5, 6, (6-3), (7-4), (6-3), (7-3), (8-4), (7-4), (8-4), (?-?), (8-4), (9-4), (9-4), (8-5), (8-5), (8-4), (9-5), (7-4), (8-5), (8-4), (7-6), (6-5), (7-4), (6-5), (6-4), 6, 5, 7, 5, 4, 5 | 7 | |
| Ind 05 (MHNCI 197) | (5-3), (5-4), 4, (3-3), (4-3), (4-3), 4, 4, 4, (3-3), 4, 3 | ? | |
| Ind 06 (MHNCI 198), ex 01 | 6, 10, 4, 10, 10, 12, (13-2), (8-3), (12-2), (9-3), 5, 7, 11, (9-4), (13-3), (14-5), (16-4), (15-5), (11-5), (9-4), (9-8), 4 | 4 | |
| Ind 06 (MHNCI 199), ex 02[1] | 5, 5, 4, 5, 5, 4, 3, 4, 5, 4 | ? | |
| Ind 07 (MHNCI 201) | 4, 3, 4, 4, 5, 4, 4, 4, 4, 4, 5, 5, 5, 5, 5, 5, 5, 5, 6, (6-4), (7-4), (7-3), (7-4), (7-4), (7-4), (7-4), (7-5), (7-5), (7-5), (7-4), (7-2), (7-5), (7-5), (6-5), (6-5), (7-5), (7-5), 5, 6, 5, 6, 5, 6, 5, 6 | 4 | |
| Ind 08 (MHNCI 202) | 2, 3, 2, 3, 3, 3, 3, 3, 3, 3, 3 | 6 | X |
| Ind 09 (MHNCI 203) | 4, 4, 4, 4, 4, 4, 4, 4, 4, 4, 4, 4, (4-2), (4-2), (5-3), (4-3), (5-3), (5-3), 5, (5-3), (4-2), (5-3), (5-3), (4-3), (5-3), (5-3), 5, 4, 5, 5, 5, 5, 4, 4, 4, 4, 5, 4, 4, 4, 4, 5, 4, 4, 4, 4, 4, 4, 4, 3, 3 | 2 | |
| Ind 10 (MHNCI 204) | ?, ?, ?, ?, ?, ?, ?, 2, 2, 3, 3, 2, 2, 3, 3, 3, 3, ?, 3, 4, 4, 3, 4, 4, 4, 4, (4-2), (4-3), 4, (4-3), (4-2), 4, (4-3), (4-1), (4-2), (4-3), (4-2), 4, (4-3), (4-3), (4-3), 5, 3, 4, 4 | 2 | |
| Ind 11 (MHNCI 205) | 1, 1, 2, 2, 2, 1, 2, 2, 2, 2, 2, 1, 3, 2, 2, 2, 2, 3, 2, 2, 3, 3, 3, 4, 3, 3, 3, 4, 3, 4, 4, 4, 4, 4, 4, 4, 4, 4, 4, 4, 4, 4, 4, 4, 4, 4, 4, 4, 4, 4, 4, 4, 4, 4, 4, 4, 4, 4, 4, 4, 4, 4, 3, 3, 4 | 2 | |
| Ind 12 (MHNCI 205) | 2, 2, 2, 2, 2, 2, 2, 1, 2, 2, 2, 2, 2, 2, 2, 2, 2, 2, 2, 3, 2, 2, 3, 3, 3, 3, 3, 3, 2, 3, 3, 3, 3, 3, 3, 3, 3, 3, 4, 2, 1 | 0 | |
| *Brachycephalus* sp. (Trilha do Corisco) | | | |
| Ind 01 (MHNCI 206) | (?-?),(?-?),(?-?),(?-?),(5-?),(?-?),(?-?),(4-?),(?-?),(?-?),(?-?) | ? | |
| Ind 02 (MHNCI 207) | 3, (4-3), (4-3), (4-3), (4-3), (4-3), (4-3), (4-3), (3-3), (3-3), (3-3), (4-3), (4-3), (3-3) | ? | |
| Ind 03 (MHNCI 208) | 4, 3, 4, 4, 4, 4, 4, 4, 4, 4, 4, 4, 4, 4, 4, 5, 5, (4-4), 5, (4-3), 4, (4-4), (5-4), (4-4), (4-3), (4-4), 4, 5, 5, 5, (5-3), 4 | ? | |
| Ind 04 (MHNCI 209) | 2, 3, 3, ?, 3, 3, 4, (3-3), (4-3), (4-3), (4-3), (3-3), (4-3), (4-3), (4-3), (3-3), (4-3), (4-3), (4-3), (3-3), (3-2), 3, (3-2) | ? | |
| Ind 05 (MHNCI 210) | 4, 4, 4, 4, 4, 4, 4, 4, (4-3), 4, (4-3), (4-3), 4, (4-3), (4-3), (4-3) | 5 | X |
| Ind 06 (MHNCI 211) | 4, (4-3), (3-3), (4-3), (4-4), (4-3), (4-4), (3-3), (4-4), (4-4), (4-4), (4-3), (4-3), (4-4), (4-4), (3-4-4), (3-4-3), (4-4), (4-4), (3-3-3), (4-3-3), (3-3-3), (3-2-3), (3-2-3), (3-3), (3-2) | ? | |
| Ind 07 (MHNCI 212) | 3, (3-2), (3-3), (3-3), (3-3), (3-3), (3-3), (3-3), (3-3), (3-3), (3-3) | ? | |

**Notes:**

[1] Only the final part of the advertisement call was recorded.

Structure of the advertisements calls (AC) recording by the author between the geographical distribution of flea toads at some point identified as *Brachycephalus sulfuratus*, *B. hermogenesi*, and as an unidentified related species, southeastern and southern Brazil. Each number represents a note, while the numerical value indicates the number of pulses for each note. Numbers in normal font outside parentheses represent isolated notes and those in normal font between parentheses represents note groups. Numbers in subscript represents attenuated notes (see text for reasons why we do not consider it as forming note groups). Question marks ("?") represents a note issued whose number of pulses could not be counted. Abbreviations: A = number of isolated notes we hear being emitted before recording the AC; B = AC emission probably interrupted due to the researcher movement in the field.
**Table 3 Parameters distinguishing the advertisement calls of flea toads at some point identified as *B. sulfuratus* and *B. hermogenesi*, including call comparisons of a third flea toad (*Brachycephalus* sp.).**

| Parameter | *B. sulfuratus* | *B. hermogenesi* | *Brachycephalus.* sp. from Corcovado and Trilha do Corisco |
|---|---|---|---|
| **Note-centered approach** | | | |
| Number of notes per call | ≤8 | ≥24 | ≥38 |
| Calls composed only by isolated notes | x | | |
| Calls present note groups | | x | x |
| Presence of warming notes | | x | x |
| Presence of attenuated notes | | x | |
| Maximum number of pulses in isolated notes | 14 | 2 | 12 |
| Maximum number of pulses per note in note groups | — | 3 | 16 |
| Maximum number of notes in note groups | — | 6[1] | 3 |
| **Call-centered approach** | | | |
| Number of notes per call | 1 | 1 | 1 |
| Calls composed only by isolated notes | — | — | — |
| Calls present note groups | — | — | — |
| Presence of warming notes | — | — | — |
| Presence of attenuated notes | — | — | — |
| Maximum number of pulses in isolated notes | — | — | — |
| Maximum number of pulses per note in note groups | — | — | — |
| Maximum number of pulses per note | 14 | 3 | 16 |
| Maximum number of notes in note groups | — | — | — |

**Notes:**
[1] Up to seven, according *Verdade et al. (2008)*.
Parameters distinguishing the advertisement calls of flea toads at some point identified as *Brachycephalus sulfuratus* and *B. hermogenesi*, including call comparisons of a third flea toad (*Brachycephalus* sp.), originally identified as *B. hermogenesi*.

to coloration in life, preventing the precise identification. Therefore, we also propose that this identification should be reverted to *Brachycephalus* sp. (being probably indeed *B. sulfuratus*; Table 1).

Advertisement calls analyzed of samples from Trilha do Corisco and Corcovado (in *part.*), two localities previously considered as occurrence of *B. hermogenesi* (*Giaretta & Sawaya, 1998*; *Verdade et al., 2008*; *Pie et al., 2013*, *2018b*; *Bornschein et al., 2016a*; *Bornschein, Pie & Teixeira, 2019*; Table 1), have reveal substantial differences to made us to considerer that represents other species, unidentified, but not *B. sulfuratus* (Tables 2 and 3). The call from this third species has two notes forming note groups, exceptionally three, and includes notes with a high number of pulses (up to 16; Tables 2 and 3). Specimens we collected at Corcovado (MHNCI 10823–5) confirmed that they belong to the *B. didactylus* species group (sensu *Pie et al., 2018b*). Three adjacent locations based on unvouchered records, Morro Cuscuzeiro. Morro do Corcovado, and municipality of Paraty (Table 1), were referred to as *Brachycephalus* sp., perhaps *Brachycephalus* sp. from Trilha do Corisco and Corcovado (Table 1; Fig. 6). This third flea toad *Brachycephalus* sp. occurs in sympatry with *B. hermogenesi* in Corcovado, as proved by our recordings (*B. hermogenesi*: MHNCI 165; *Brachycehalus* sp.: MHNCI 165–205). The phylogenetic

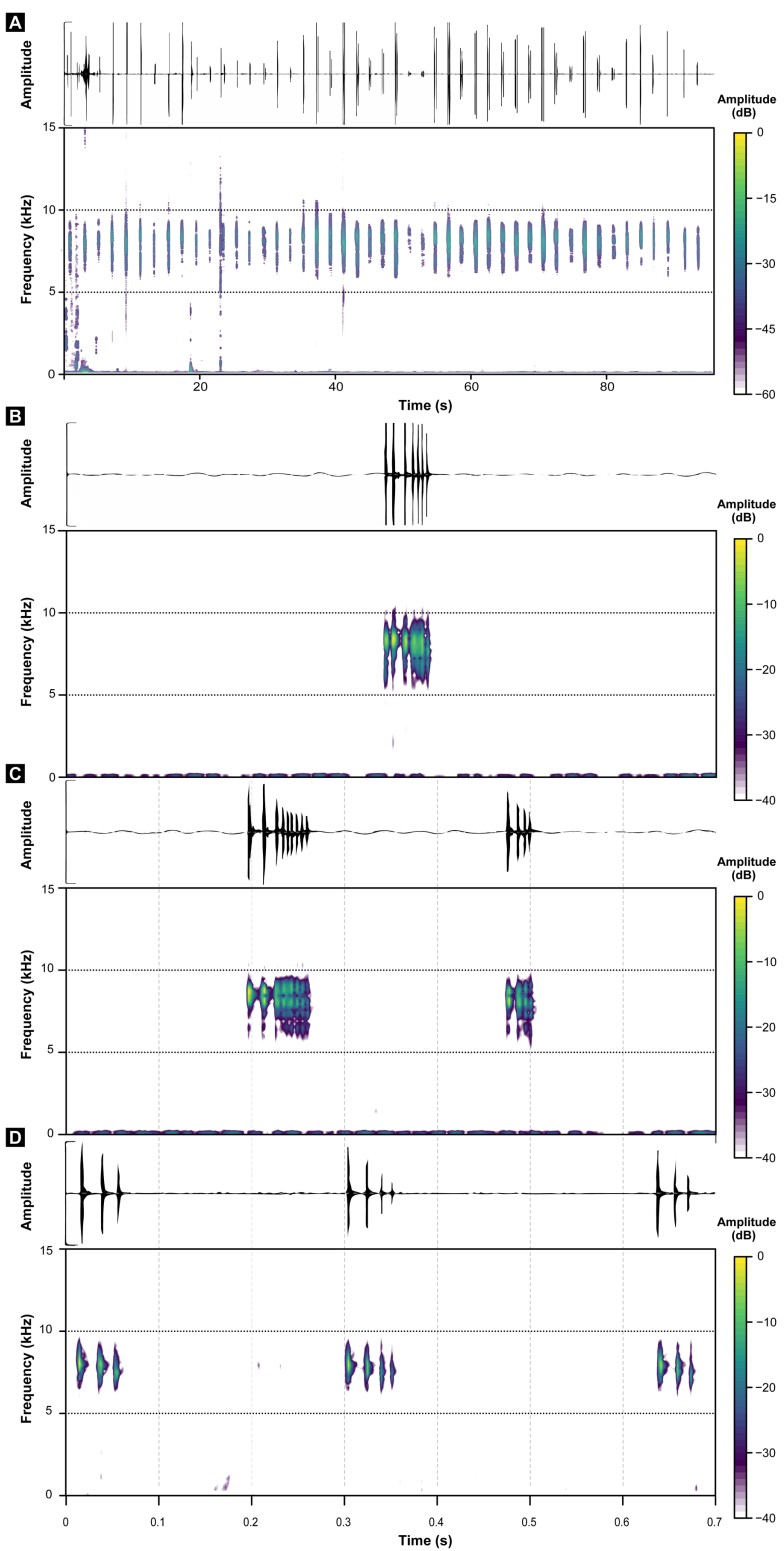

**Figure 5 Oscillograms and spectrograms of *Brachycephalus* sp. (other than *B. sulfuratu*s and *B. hermogenesi*).** (A) Example of one entire call with 71 notes recorded (MHNCI 200; Corcovado, municipality of Ubatuba, São Paulo; M. R. Bornschein). (B) Example of one isolated note with seven pulses (MHNCI 198; Corcovado; M. R. Bornschein). (C) Example of one note group with two notes (with

**Figure 5** (continued)
nine and four pulses, respectively; MHNCI 198). (D) Example of one note group with three notes (the first note with three pulses and the remaining notes with four pulses; MHNCI 211; Trilha do Corisco, municipality of Paraty, Rio de Janeiro; L. F. Ribeiro). Spectrograms are produced with Hann window, overlap of 50%, and FFT size of 16,384 points in (A) and 256 points in (B)–(D).

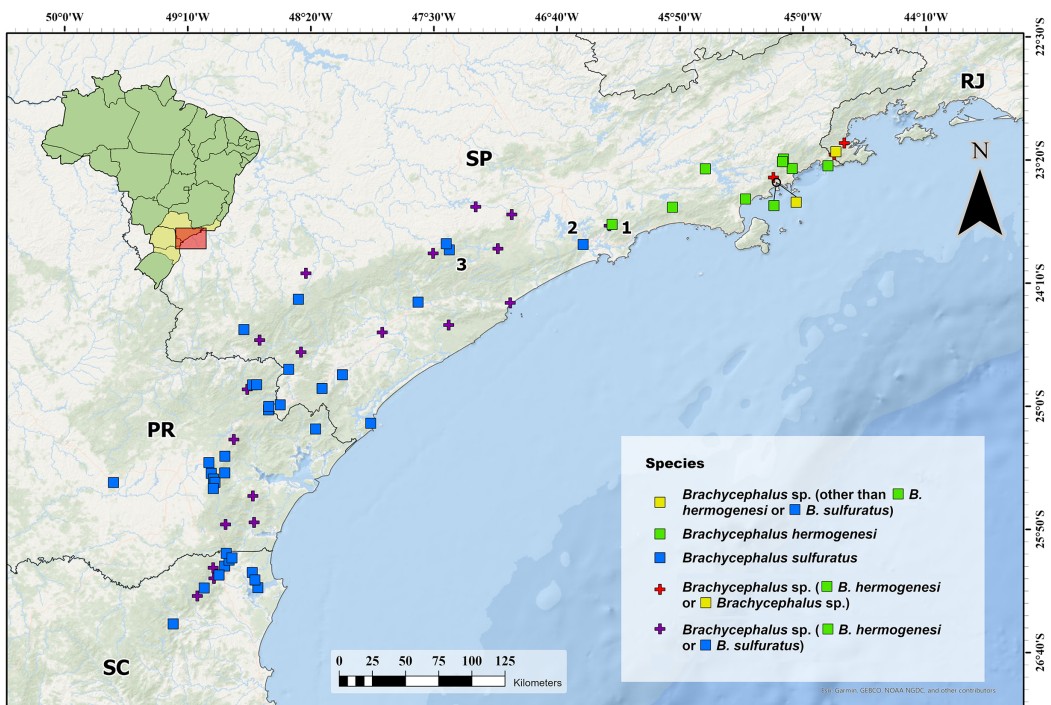

**Figure 6 Current identification of records of flea toads that have been at some point identified as *Brachycephalus sulfuratus*, *B. hermogenesi*, and as an unidentified related species.** Current identification of records of flea toads that have been at some point identified as *Brachycephalus sulfuratus*, *B. hermogenesi*, and as an unidentified related species, according to the compilation of localities and review of identifications shown in Table 1. We highlighted the southernmost record of *B. hermogenesi* confirmed (1—Parque Natural Municipal Nascentes de Paranapiacaba). We also highlight the northernmost confirmed records of *B. sulfuratus* (2—Núcleo Itutinga-Pilões and 3—near the Jurupará dam). Abbreviations: RJ = Rio de Janeiro; SP = São Paulo; PR = Paraná; SC = Santa Catarina. Map image is the intellectual property of Esri and is used herein under license. Copyright © 2020 Esri and its licensors. All rights reserved.

analysis revealed that the specimen from Municipality of Paraibuna is indeed *B. hermogenesi* (Table 1), being placed with other specimens of the species collected at the type locality (Fig. 7).

## DISCUSSION

Based on our analyses, characters previously used as diagnostic for *B. sulfuratus* were quite variable and overlapped with those of *B. hermogenesi*. Moreover, the examination of specimens deposited in the collections MHNCI and ZUEC support this claim. Currently, differences in the call structure—number of notes per call and presence/absence of note

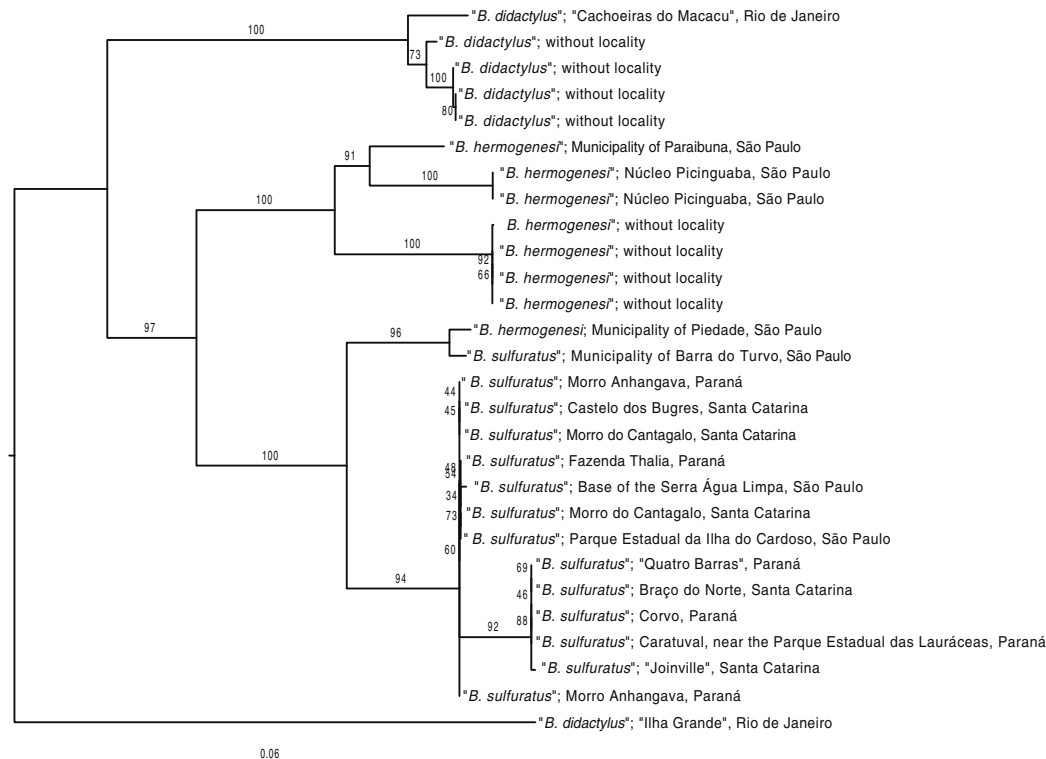

**Figure 7 Phylogenetic tree based on a concatenated dataset of all mitochondrial 12S and 16S mitochondrial loci available on GenBank for specimens of the *B. didactylus* species group.** Phylogenetic tree based on a concatenated dataset of all mitochondrial 12S and 16S mitochondrial loci available on GenBank for specimens of the *B. didactylus* species group (Table S1). The tree was rooted by its midpoint. Whenever possible, the corresponding localities available on their GenBank records were standardized based on the toponyms indicated in Table 1. Notice that the specimen originally identified as *B. hermogenesi* from the Municipality of Piedade (*Condez, Sawaya & Dixo, 2009*, *Clemente-Carvalho et al., 2011*), was reverted to *B. sulfuratus* (Table 1). Branch values correspond to bootstrap support.

groups - is proposed here as the only available sources of evidence supporting the distinction between *B. sulfuratus* and *B. hermogenesi*. Even in the field its advertisement calls are very distinct to the human ear and easily distinguishable. The advertisement calls of *B. sulfuratus* sounds like a "tríííííí, tríííííí, tríííííí, tríííííí, tríííí", whereas the calls of *B. hermogenesi* from it type locality sound like a "tíc, tíc, tíc, tíc-tíc, tíc-tíc-tíc, tíc-tíc-tíc, tíc-tíc-tíc, …". These transliterations represent isolated notes or note groups (each note separated by comma and note group by hyphen) with distinct durations (= transliteration size) related to the number of notes in the call. This diagnosis between *B. sulfuratus* and *B. hermogenesi* is only feasible under the note-centered approach. Considering their calls under the call-centered approach, there would be no diagnosis to be proposed between them at this moment, because each note would represent a call (Table 3). To the best of our knowledge, this is the first case in which the diagnosis between species of any *Brachycephalus* is made solely by characters of their advertisement call.
The first notes emitted from an advertising call by *B. hermogenesi* are usually hardly noticed in the recording and equally difficult to hear in the field. This is the reason why we rarely record the first emissions and many recordings recorded the advertisement call already in progress. These weak starting notes of an advertisement call were called warming notes (*Bornschein et al., 2018*; Table 3), assuming that they would reflect the individual's preparation process to the level of excitement required for the issuance of "typical" strongest notes. Like warming notes, attenuated notes could prepare the individual to issue the immediately subsequent notes at a higher level of arousal.

The recognition of the existence of warming notes and attenuated notes, as well as the existence of note groups for understanding the richness of characters in *Brachycephalus* calls (see also above), consolidate the benefit of the note-centered approach over the call-centered approach in describing calls of species of this genus (*Bornschein et al., 2018*). The note-centered approach way for description the calls of *B. hermogenesi* also reinforces the hypothesis of complexity increment along note emissions (*Bornschein et al., 2018*), with the incorporation of note groups during the call emission. These structural particularities would not be perceived under the call-center approach. Under this approach, they would be perceived as a simple intraspecific variation in calls

The advertisement calls of *B. hermogenesi* show the same pattern as species from the *B. pernix* group (*Bornschein et al., 2018*, *2019*; *Pie et al., 2018b*; *Monteiro et al., 2018b*, *2018a*), which includes most species of southern Brazil, whereas the call of *B. sulfuratus* resembles the call of *B. vertebralis* (MRB, unpublish data), for example, from the *B. ephippium* group, which includes most species from the state of São Paulo to the north up to Espírito Santo and Minas Gerais.

We now confirm the absence of occurrence records of *B. hermogenesi* in southern Brazil and the presence of *B. sulfuratus* as far north as the east of São Paulo city, only 25 km in straight line from the southernmost site of a confirmed record of *B. hermogenesi* (Parque Natural Municipal Nascentes de Paranapiacaba; Fig.6; Table 1). Most unidentified records (Table 1) represent one or the other of these two species. In fact, it is likely that in southern Brazil only the flea toad *B. sulfuratus* occurs. In this region, our research group has been working with two anuran genera (*Brachycephalus* and *Melanophryniscus*) since 2009, focusing on their distribution, ecology and conservation (*Pie et al., 2013*; *Bornschein et al., 2015*, *2016a*; *Bornschein, Pie & Teixeira, 2019*), and thus are particularly aware of *Brachycephalus* calls wherever we do field work and yet we never recorded *B. hermogenesi* calls in southern Brazil.

In addition, we also underscore the absence of records of *B. sulfuratus* in northern Santa Catarina in some well sampled localities. For example, we obtained no records for *B. sulfuratus* in Morro Boa Vista (26°30′58″S, 49°03′14″W), on the border between the municipalities of Jaraguá do Sul and Massaranduba, where we described *B. albolineatus* (*Bornschein et al., 2016b*), Morro do Baú (26°47′58″S, 48°55′47″W), municipality of Ilhota and Morro Braço da Onça (26°44′58″S, 48°55′41″W), municipality of Luiz Alves, where we report *B. fuscolineatus* (*Ribeiro et al., 2015*; *Bornschein, Teixeira & Ribeiro, 2019*), Morro do Cachorro (26°46′42″S, 49°01′57″W), on the border between the municipalities of Blumenau, Gaspar, and Luiz Alves, where we described *B. boticario*

(*Ribeiro et al., 2015*), and Morro Santo Anjo (26°37′41″S, 48°55′50″W), municipality of Massaranduba, where we described *B. mirissimus* (*Pie et al., 2018b*). It is possible that the southern limit of the geographical distribution of *B. sulfuratus* occurs at the Morro do Garrafão (Table 1).

Contrary to what is found in southern Brazil, the distribution of flea toads in the states of São Paul and southern Rio de Janeiro are poorly known. Our findings indicate the presence of a third flea toad species at the border between São Paulo and Rio de Janeiro states, at least occurring in Corcovado, São Paulo, and Trilha do Corisco, municipality of Paraty, Rio de Janeiro. Corcovado, however, is one locality of paratypes of *B. hermogenesi* and Paraty were also cited as a place of occurrence of *B. hermogenesi* in the original description of this species (*Giaretta & Sawaya, 1998*). The species of the *B. didactylus* group that occurs closest to Rio de Janeiro/São Paulo border, excluding *B. hermogenesi* and *B. sulfuratus*, is *B. didactylus*, in Vila Dois Rios, Ilha Grande, municipality of Angra dos Reis, Rio de Janeiro (*Bornschein, Pie & Teixeira, 2019*). The Trilha do Corisco is distant from Vila Dois Rios 59 km in a straight line.

As we demonstrate in our analyses, there is no confirmed overlap in the distribution of *B. hermogenesi* and *B. sulfuratus*, and their geographical replacement occurs in southeastern of São Paulo city, without apparent barriers. There are other examples of discontinuity of the geographical distribution between congeneric species throughout the Atlantic Forest from southeastern to southern Brazil in southeastern São Paulo city, as in the montane bird *Scytalopus speluncae* (taxonomy sensu *Maurício et al. (2010)*). *Maurício (2005)* stated that populations of *S. speluncae* from the southeastern of the city of São Paulo to the south of the species distribution represent a distinct species yet to be named, and he treated it as "Southern *Scytalopus speluncae*" (this scenario of southern population of this bird as a new species was supported by other studies (*Bornschein et al., 2007*; *Mata et al., 2009*; *Maurício et al., 2014*; *Pulido-Santacruz et al., 2016*)). In the region around the southeastern of São Paulo city, cases of hybridization of subspecies or lineages have been reported for at least four species of birds (*Pinto, 1941*; *Silva & Stotz, 1992*; *Cabanne, Santos & Miyaki, 2007*; *D'horta et al., 2011*; see also *Dantas et al., 2015*). In the state of São Paulo there is another discontinuity which is associated with intraspecific differentiation or even sister species of frogs (*Fitzpatrick et al., 2009*; *Thomé et al., 2010*; *Amaro et al., 2012*) and snakes (*Grazziotin et al., 2006*).

The correspondence between the distribution of the congeneric species in question with the limits of the Serra do Mar is intriguing, given that during the last 20 million years there was no obvious uplift in the region (*Gontijo-Pascutti et al., 2012*). This time scale is considerably older than the inferred cladogenesis events and therefore geological processes could not have been the primary cause of their divergence, given that *Brachycephalus* toads and *Scytalopus* birds of São Paulo, Paraná, and Santa Catarina originated less than 2–5 million years ago (*Pie et al., 2018a* and *Pulido-Santacruz et al., 2016*, respectively). Likewise, recent neotectonic activities (Late Pleistocene-Holocene) are restricted to the faults and stress regimes (*Hasui, 1990*; *Saadi, 1993*; *Ricommini & Assumpção, 1999*) and, therefore, also could not have generated the diversification pattern of widely distributed terrestrial species. It is important to note that *Thomé et al. (2010)*, studying the toad

*Rhinella crucifer* from the eastern portion of Brazil, associate one genetic break found in eastern São Paulo to neotectonic barriers, specifically the Cubatão shear zone and the Guapiara lineament. However, these are ancient geotectonic activities, from Proterozoic to Cambrian (with Phanerozoic reactivation) and Mesozoic, respectively (*Ferreira et al., 1981*; *Sadowski, 1991*; *Almeida & Carneiro, 1998*; see also *Ricommini & Assumpção, 1999*). In addition, studies have proposed speciation by vicariance caused by relatively recent events, such as river barriers (*Amaral et al., 2013*), sea level variation (*Grazziotin et al., 2006*; *Fitzpatrick et al., 2009*), and forest refugia (*Fitzpatrick et al., 2009*; *Thomé et al., 2010*; *D'horta et al., 2011*; *Amaral et al., 2013*). The largest river around the disruption of the geographical distribution of *B. sulfuratus* and *B. hermogenesi* is the Rio Ribeira do Iguape, which intersects the Serra do Mar between São Paulo and Paraná States by continued erosive retreat (*Almeida & Carneiro, 1998*). Alternatively, the disruption of the Serra do Mar in that region originated from a tectonic depression associated with the asymmetric graben of the Sete Barras or Ribeira de Iguape (*Melo et al., 1989*; *Gontijo-Pascutti et al., 2012*). However, the formation of the present configuration of the Serra do Mar did not lead to isolation, given that *B. sulfuratus* occurs on both banks of the Ribeira do Iguape river. It is plausible that the origin of *B. sulfuratus* and *B. hermogenesi*, as well as the other examples mentioned above, might have resulted from climatic variations that promoted vicariance by forest cover disruption followed by the recovery of forest cover, presumably leading to secondary contact.

The region in the state of São Paulo, around the southeastern São Paulo city, should be further investigated. Records of flea toads in this region could be obtained as background sound in recordings of birds (e.g., recordings deposited in databases such as www.xeno-canto.org and www.wikiaves.com.br; Table 1). *Verdade et al. (2008)* made a similar suggestion: to search for records of *B. hermogenesi* in the background of recordings of birds from the Estação Biológica de Boracéia, in the case one wants to seek previous records of this flea toad in this highly sampled locality. As examples, calls of *B. sulfuratus* in Parque Estadual Intervales, municipality of Iporanga, state of São Paulo (Table 1), can be heard in recordings of the birds *Merulaxis ater* (XC80463 and XC18179) and *Eleoscytalopus indigoticus* (XC75544; available at www.xeno-canto.org), and calls of *B. hermogenesi* in Núcleo Santa Virgínea, Parque Estadual da Serra do Mar, municipality of São Luiz do Paraitinga, São Paulo, can be heard in a recording of *E. indigoticus* (XC253045; Table 1).

We underscore the importance of continuous scrutiny of the distribution and advertisement call analysis of *B. sulfuratus* and *B. hermogenesi*. The advertisement calls of *B. hermogenesi* need to be redescribed (see *Pie et al., 2018b*: 12) and a better understanding of the geographical limits between this species and *B. sulfuratus* can elucidate distribution patterns and potentially detect cases of sympatry. To date, the occurrence of *B. hermogenesi* and *Brachycephalus* sp. (other than *B. sulfuratus* and *B. hermogenesi*) at Corcovado, São Paulo, is the only confirmed case of sympatry between species of *Brachycephalus* in the same group. Other cases of sympatry include *Brachycephalus* from distinct groups (*B. pernix* and *B. didactylus* groups and *B. ephippium*

and *B. didactylus* groups; *Bornschein et al. (2016a)* and *Bornschein, Pie & Teixeira (2019)*). Even in sympatry, the differences between the calls of *B. hermogenesi* and *Brachycephalus* sp. and between *B. hermogenesi* and *B. sulfuratus* are substantial and could provide pre-zygotic isolation. Although some species in the *B. ephippium* group are additively insensitive to the own advertisement call (*Goutte et al., 2017*), which would suggest loss of active selection pressure and variation maintained by inertia, it must be considered that this scenario may not apply to the other groups (*Monteiro et al., 2018b*) and, also, that the species may actively perceive call emissions through vibrations in other body receptors.

## CONCLUSIONS

*Brachycephalus sulfuratus* differs from *B. hermogenesi* only by its advertisement calls; other morphological characters previously suggested to distinguished *B. sulfuratus* from *B. hermogenesi* are extremely variable and show overlap between these two species. The advertisement calls of these species differ greatly from each other and can be easily recognized by the human ear in the field. *Brachycephalus sulfuratus* presents few notes per call with only isolated notes and *B. hermogenesi* present high number of notes per call with isolated notes and note groups. The advertisement calls of *B. sulfuratus* resemble those of species of the *B. ephippium* species group, whereas the calls of *B. hermogenesi* resemble those of the *B. pernix* species group. Understanding the evolution of these advertisement calls should require a more in-depth investigation.

All previous records of *B. hermogenesi* from southern Brazil should instead be considered as *B. sulfuratus*, in a possibly cascading error resulting from the inadequate revision of the records prior to the description of *B. sulfuratus* (*Condez et al., 2016*). A large region in the south of the state of São Paulo needs to be further investigated to confirm the presence of *B. hermogenesi*; the previous records were reverted to *Brachycephalus* sp. *Brachycephalus sulfuratus* is distributed much further north than previously thought and it is possible that sympatry with *B. hermogenesi* may occur in the southeast of the city of São Paulo. This region in the southeast of São Paulo is particularly interesting because many species of different taxa have their range limits there. The biogeographic explanation of this pattern seems to be limited to the past distribution of forest patches, which could have been previously isolated and are now distributed continuously, allowing possible secondary contact of species.

The *B. hermogenesi* type series possibly includes a second species of flea toad, not yet identified. This situation involves a locality of a *B. hermogenesi* paratype, and probably not the holotype. Therefore, there is no evidence, at this moment, to suspect the name *B. hermogenesi* as a possible synonym for another *Brachycephalus* species, as *B. didactylus*, for example. It is necessary to deepen the field studies to identify the local populations and to clarify the limits of the geographic distribution, as well as to review the identification of museum material, including the type series of *B. hermogenesi*.

Phylogenetic analysis provided evidence that at least *B. sulfuratus* probably includes more than one species under this name, although this species, as presently defined, has a similar calling pattern in its wide geographical distribution, from southeastern São Paulo to Santa Catarina (Table 1; Fig. 6). In parallel, our *B. hermogenesi* call analyses provided the

first association of a call pattern across the geographic distribution of this species (Table 1; Fig. 6), but this does not mean that only one species is necessarily included under this name, because distinct species of *Brachycephalus* may have indistinct calls (*Pie et al., 2018b*). Combined with the fact that the *B. didactylus* group includes cryptic species, difficult or even impossible to identify in preservative, that occur or can occur locally in sympatry, we recommend a solid and broad review of the taxonomy of the group based on own analyses of large series of specimens and calls.

## APPENDIX 1

Advertisement calls analyzed in the present study. Abbreviation: MHNCI = Museu de História Natural Capão da Imbuia, Curitiba, Paraná.

*Brachycephalus sulfuratus*. SÃO PAULO: Base of the Serra Água Limpa, municipality of Apiaí MHNCI 129; Biquinha, municipality of Juquiá MHNCI 128; near the Jurupará dam, municipality of Piedade MHNCI 123–5; Núcleo Itutinga-Pilões, Parque Estadual da Serra do Mar, municipality of Cubatão MHNCI 126–7; Serra do Guaraú, on the border of the municipalities of Cajati and Jacupiranga MHNCI 130; Torre Embratel, municipality of Cajati MHNCI 218. PARANÁ: Caratuval, near the Parque Estadual das Lauráceas, municipality of Adrianópolis MHNCI 131; Caratuval, Parque Estadual das Lauráceas, municipality of Adrianópolis MHNCI 132; Entroncamento Teba, Rio Turvo, municipality of Campina Grande do Sul MHNCI 219; Fazenda Thalia, municipality of Balsa Nova MHNCI 134; Morro do Canal, municipality of Piraquara MHNCI 220; Reserva Particular do Patrimônio Natural Salto Morato, municipality of Guaraqueçaba MHNCI 133. SANTA CATARINA: Monte Crista, municipality of Garuva MHNCI 221; Morro do Garrafão, municipality of Corupá MHNCI 137; Morro Garuva, municipality of Garuva MHNCI 136; Serra do Pico, municipality of Joinville MHNCI 217; Truticultura, municipality of Garuva MHNCI 135.

*Brachycephalus hermogenesi*. SÃO PAULO: Corcovado, municipality of Ubatuba MHNCI 166; Estação Biológica de Boracéia, municipality of Salesópolis MHNCI 166–9; Morro do Cantagalo, municipality of Caraguatatuba MHNCI 222–3; Núcleo Cunha, Parque Estadual da Serra do Mar, municipality of Cunha MHNCI 170–1; Núcleo Picinguaba, Parque Estadual da Serra do Mar, municipality of Ubatuba MHNCI 172–87; Parque Natural Municipal Nascentes de Paranapiacaba, municipality of Santo André MHNCI 213–6; Trilha do Ipiranga 50 m from the Rio Ipiranga, Núcleo Santa Virgínia, Parque Estadual da Serra do Mar, municipality of São Luiz do Paraitinga MHNCI 188–92.

*Brachycephalus* sp. (other than *B. sulfuratu*s and *B. hermogenesi*). RIO DE JANEIRO: Trilha do Corisco, municipality of Paraty MHNCI 206–12. SÃO PAULO: Corcovado, municipality of Ubatuba MHNCI 193–205.

## ACKNOWLEDGEMENTS

Diego Baldo, Stefano Spiteri, and André Confetti provided valuable assistance during field work. Vanessa K. Verdade provided two samples of the advertisement call of *Brachycephalus hermogenesi*. Milene Fornari provided bibliography for the discussion on

biogeography. We thank three anonymous reviewers for valuable comments on the manuscript.

### Funding

Fieldwork from 2011 to 2019 was partially funded by Fundação Grupo Boticário de Proteção à Natureza (through grants 0895_2011, A0010_2014 and 1149_20191). Other fieldworks from 2018 to 2019 was funded by National Geographic Society (through the grant EC–50722R-18 to Larissa Teixeira). Marcio R. Pie was supported through a grant from CNPq/MCT (571334/2008–3). There was no additional external funding received for this study. The funders had no role in study design, data collection and analysis, decision to publish, or preparation of the manuscript.

### Grant Disclosures

The following grant information was disclosed by the authors:
Fundação Grupo Boticário de Proteção à Natureza: 0895_2011, A0010_2014, and 1149_20191.
National Geographic Society: EC–50722R-18.
CNPq/MCT: 571334/2008–3.

### Competing Interests

Marcio R. Pie is an Academic Editor for PeerJ.

### Author Contributions

- Marcos R. Bornschein conceived and designed the experiments, performed the experiments, analyzed the data, prepared figures and/or tables, authored or reviewed drafts of the paper, and approved the final draft.
- Luiz Fernando Ribeiro analyzed the data, prepared figures and/or tables, **carried out field work,** and approved the final draft.
- Larissa Teixeira analyzed the data, prepared figures and/or tables, and approved the final draft.
- Ricardo Belmonte-Lopes analyzed the data, authored or reviewed drafts of the paper, fieldwork and digitalization of recordings, and approved the final draft.
- Leonardo Amaral de Moraes analyzed the data, prepared figures and/or tables, authored or reviewed drafts of the paper, and approved the final draft.
- Leandro Corrêa analyzed the data, authored or reviewed drafts of the paper, field work, and approved the final draft.
- Giovanni Nachtigall Maurício analyzed the data, authored or reviewed drafts of the paper, and approved the final draft.
- Júnior Nadaline analyzed the data, authored or reviewed drafts of the paper, field work, and approved the final draft.
- Marcio R. Pie analyzed the data, authored or reviewed drafts of the paper, and approved the final draft.

## Animal Ethics

The following information was supplied relating to ethical approvals (i.e., approving body and any reference numbers):

Collection permits for this study were issued by ICMBIO (10.500, 22470–2/1911426, and 55918–1).

## Data Availability

Raw recording data is available in Table 2.

All specimens are deposited in the collection of the Museu de História Natural Capão da Imbuia, Curitiba, Paraná, Brazil (MHNCI 123-137, MHNCI 165-223).

Recordings are available in the Supplemental Files.

## Supplemental Information

Supplemental information for this article can be found online at http://dx.doi.org/10.7717/peerj.10983#supplemental-information.

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
