# Peer review of "A review of the diagnosis and geographical distribution of the recently described flea toad Brachycephalus sulfuratus in relation to B. hermogenesi (Anura: Brachycephalidae)"

_PeerJ, doi:10.7717/peerj.10983_

## Round 0.1 · original submission · Major Revisions

Thank you for submitting your work to PeerJ. I have sent your paper to the three expert referees for their consideration. I have now received their comments back and have read through your paper carefully myself. Enclosed please find the reviews of your manuscript. All of us agree that the manuscript can be considered for publication after a major revision. The reviewers find merit in your work, however they raise a number of important questions and concerns which have to be addressed prior to acceptance of your paper to publication. Some of these issues are critical for interpreting your data and taxonomic results. Therefore I would ask you to revise your manuscript in accordance with suggestions of the reviewers. Please address all the questions raised on a point by point basis. After I get the revised version of your paper it will be forwarded to the same referees for consideration.

Reviewer 1 ·

Basic reporting

The article could benefit from a review by a native speaker.

Literature is comprehensive.

The structure of the article could be improved.

Apparently, the raw data is not shared.

The article is not framed in the formal context of hypotheses, assumptions, and predictions, but is organized in terms of clear and well-defined questions.

Please, see "General comments for the author".

Experimental design

The research question is well defined, relevant, and meaningful. The gap in knowledge is identified precisely. The technical standards adhere to current practice in taxonomy. Methods are well described so as to allow replication.

Validity of the findings

The findings are valid are absolutely relevant to the problem at hand.

Additional comments

Title

The title reads "A review of the geographical distribution and differentiation of the recently described flea toad Brachycephalus sulfuratus in relation to B. hermogenesi (Anura: Brachycephalidae)"

The word "differentiation" implies a process. Perhaps the Authors really mean "diagnosis"? Perhaps the logical order of formulating the problem could be "diagnosis and geographical distribution" instead of "geographical distribution and differentiation"?


Introduction

The problem of interest is species diagnosis and geographical distribution. The problem does not seem, however, to be clearly and logically stated in the Introduction. Perhaps, the Authors could formulate the problem from the general to the specific in three steps. First, the definition of the species groups. Second, define the species group of interest and the component species. Third, define the question of diagnosis and distribution of the target species. Furthermore, the Authors waste time and text surmising what Condez et al. might or might not have known or done. This renders the Introduction speculative and unduly long, besides distracting from the logic of the scientific problem. The Introduction could be shortened considerably and made more precise if the Authors concentrated on the fundamental problem of the diagnosis and distribution of the organism of interest.

Page 9; line 86: "Ommit" is defined as "to leave out or exclude (someone or something), either intentionally or forgetfully". The Authors do not actually know whether Condez et al. ommited the work of Pie et al. The Authors also do not know whether Condez et al. have an "opinion" about the matter. In fact, this seems to be a "opinion" of the Authors. Perhaps the Authors could reconsider their parlance, which might in fact betray a preconceived notion they may have about Condez et al.´s work. Perhaps the Authors could simply state that Condez et al. did not take into account the information available in Pie et al.

Page 9; lines 94 - 97: The meaning and significance of this sentence is difficult to grasp. What information are the Authors trying to convey? Do the Authors mean that, apparently, three different species occur in a single locality? What is the difference between Brachycephalus sp. 1 and Brachycephalus sp. nov. 1?

Page 9; lines 97 - 98: The Authors do not know why Condez et al. did not examine specimens from Castelo dos Bugres. Therefore, the Authors cannot ascertain whether Condez et al. "... overlooked specimens from this locality ..." and "... missed the opportunity …”. Perhaps the Authors might consider revising their own reasoning and, consequently, their parlance.

Page 9; lines 98 - 100: Is the "uncertainty" related to the occurrence, that is, the geographical location or to the species diagnosis?


Material & Methods

Page 10; line 132: “… with a few updates”. This statement is devoid of information. Please, be specific.

Page 10; line 132: What is the meaning of “… parameters …”?

Page 10; lines 132 - 134: “We looked for those features …”. Perhaps the use of demonstrative adjectives such as “those” should be avoided in scientific writing because such adjectives may render the text ambiguous. To what features are the Authors referring to? In this connection, the Authors use five words, namely, attribute, character, characteristic, feature, and trait. For the sake of scientific clarity, might the Authors be consistent and use one of these words throughout the text?


Results

What are exactly the new traits that the Authors uncovered as useful to discriminate between the species of interest?


Discussion

Pages 17 - 18; lines 352 – 388: The fundamental problems of diagnosis and geographic distribution are not yet fully resolved for the organisms of interest. Therefore, the biogeographical and evolutionary illations lack in foundation and, in fact, do not contribute to the primary task of resolving the problem of diagnosis itself. Perhaps the Authors might consider deleting this section of the text.


Final comment.

The manuscript is absolutely relevant and does contribute valuable information on the taxonomy of Brachycephalus. The work has been carried out by a research group that has consistently produced first rate work on Brachycephalus. The manuscript could be considerably streamlined to make the argument clear, concise, and precise. Furthermore, the manuscript would greatly benefit from a review by a native speaker.

Reviewer 2 ·

Basic reporting

I found the manuscript quite confusing. There were several sentences I had trouble following (e.g. lines 128–143; 167–176). Some sentences lack structure and streamlining (e.g. lines 132–133), and overall, the manuscript was a bit wordy and the grammar and usage of English clearly need to be improved. I think detailed copyediting would be helpful. I believe that the authors should be careful with the diagnosis concept.
Literature references and backgrounds are good.
A comparative table is required. Including morphological and bioacoustical features for B. hermogenesi and B. sulfuratus.

Experimental design

The material and methods should be improved. There is not enough information.
Specimens examined list are required.

Validity of the findings

The results are not robust and do not support the conclusion.

Additional comments

Here, I return my review of the manuscript entitled: “A review of the geographical distribution and differentiation of the recently described flea toad Brachycephalus sulfuratus in relation to B. hermogenesi (Anura: Brachycephalidae)”, by Bornschein et al. The manuscript should be accepted with major review and go through another round of revision. The authors persuade me that the diagnosis provided by Condez et al. (2016) is not enough to distinguish between B. hermogenesi and B. sulfuratus. However, currently, it is beyond the manuscript scope. In my opinion, the authors have two options to improve the manuscript, and in both of them, they should examine voucher specimens from all localities provided in Figure 7. 1) Stick with the present title, rebuild the goals, remove “Distinction between Brachycephalus sulfuratus and B. hermogenesi” part from the results and rewrite the discussion and conclusion focusing in the geographical part. 2) The way the manuscript is written they have no evidence to go beyond. But they might choose another option, and include a diagnosis for each species, and go in-deep into it. You also have to analyze specimens of B. hermogenesi, even in the advertisement calls. Then, see all my comments throughout the manuscript and start modifying the title.
I believe that the authors should be careful with the diagnosis concept. A species can be diagnosed by one or several exclusive features (as was done by Giaretta and Sawaya 1998 – e.g. presence of a functional fifth toe) or by a combination of characters as was done by Condez et al. (2016). Furthermore, the results should contain the author's observations, in their study, and it not happened. The way that part of the discussion, which is in the results, and part of the discussion is described seems like gossip. What do the authors really mean? Try to be direct. That way the reading is boring.
Introduction: the authors criticize the description of B. sulfuratus by Condez et al. (2016) saying that the authors overlooked previous identifications of Brachycephalus sp. nov. and B. hermogenesi. However, several manuscripts cited in the introduction of the present study cite manuscripts that do not include or examine any specimens and are a compilation of species distribution data. In my opinion, they should rewrite the introduction, otherwise seems that they are just angry with Condez et al. (2016). In this sense, I strongly suggest the authors rewrite the introduction focusing on the morphological, bioacoustical, and geographical overlap. Additionally, you can include information as: “that are still unresolved issues regarding species-level identification in the recent literature (cite all literature with Brachycephalus sp.) several specimens identified as putative new species (all identified as Brachycephalus sp. nov. 1)”.
Material and Methods: you must provide a number of specimens per species and locality (for both morphological and bioacoustical data), otherwise, this manuscript has no value. The authors have to provide a paragraph with fieldwork information for recordings and to say where their species (as they refer to the specimens) were collected. Furthermore, they not provide any information on the terminology of recordings.
Results: you should begin with the diagnosis of each species, including morphological and bioacoustical features. Then, you should compare B. sulfuratus and B. hermogenesi and finally. For each described feature you must provide the number of analyzed specimens that have that state of the feature, as well as the percentage of individuals. In this way, we have an idea of how many individuals you analyzed and how many of them have that state of that feature. Would be a good idea to provide a comparative morphological table (between B. hermogenesi and B. sulfuratus). The “Reviewed records of Brachycephalus sulfuratus and B. hermogenesi” is the best part of your manuscript.
Discussion: It would be an excellent idea to include the diagnostic list from Condez et al. (2016), improving that with the observed features in your study (provide numbers, and say what numbers were from Condez et al. 2016 and what numbers are your result). Also, do this with B. hermogenesi.

Annotated reviews are not available for download in order to protect the identity of reviewers who chose to remain anonymous.

Reviewer 3 ·

Basic reporting

Marcos Bornschein and his colleagues presented a manuscript clarifying the identification of Brachycephalus sulfuratus and revising published and unpublished records of B. sulfuratus, B. hermogenesi, and the third taxon with a dubious specific status. The manuscript is well written, and its results improve the taxonomical knowledge by identifying an acoustic character as the only diagnostic between these two cryptic species, as well as improve the knowledge on their geographic distribution. The authors have a powerful dataset in hands, but I feel it could be better explored, mainly the acoustic data; see my comments in the next topic “Experimental design”.

I am not a native English speaker, but I feel the English language is clear in most parts. However, I recommend authors sending the manuscript to revision by a native after the manuscript revision to ensure that the language is clear to all the audience (including native English speakers). The provided literature references and the field background is in accordance with the manuscript’s goal.

Experimental design

Materials and Methods are poorly informative, especially about acoustic sampling. In the current form, the study is not even close to being replicable. Authors must provide detailed information on how many specimens they analyzed, how many calls they have (how many they collected and where they collected; how many recordings are from public database, etc). I recommend them to include a summary of this information in the text, and then, a table with localities where they collect the specimens and recordings.

Validity of the findings

The findings are very relevant but could be stronger than they currently are.

(1) The authors have a strong acoustic dataset, which could allow them to measure some acoustic parameters of the several recordings for both species they have in hand (not only visually check the sonograms in order to find "visual" differences [the number of pulses in single notes]), especially the temporal parameters. Authors stated in the discussion that measuring the acoustic traits was not the "purpose for which calls are included" in the study, but the absence of this data makes the manuscript less strong than it has the potential to be. Sounds like authors are splitting data into several manuscripts instead of publishing their evidence in a robust manuscript. I strongly recommend authors to analyse their acoustic records for both B. sulfuratus, B. hermogenesi. They could use the data to describe, in detail, the advertisement call of this species. This kind of result would help (a lot) other researchers studying these small toads. Furthermore, other diagnostic characters could arise from these results (and it is clearly the purpose of the present study).
(2) Similarly to the acoustic dataset, authors have also several distribution records of both species. Authors discussed the geographic distribution of B. sulfuratus and B. hermogenesi, and raised a hypothesis to explain the allopatric distribution of these species in the absence of any evident geographic barrier in the state of São Paulo. Currently, the discussion on this issue is very speculative. To change this scenario, authors could run a distribution model for both species using present/past climate variables. It would allow the authors to discuss possible factors driving the distribution of these species, as well as what may have promoted the origin of these species.

Additional comments

All comments were given above. Se the commented PDF for minor issues or suggestions.

Annotated reviews are not available for download in order to protect the identity of reviewers who chose to remain anonymous.

---

## Round 0.2 · accepted · Accept

Thank you for taking the time to revise and resubmit your manuscript. I have now read through your paper as well as your letter in response to the reviews. I think - and both reviewers agree - that you have successfully addressed all of the concerns raised very well, and would like to accept your manuscript for publication in PeerJ. Congratulations!

Please note that the second reviewer has raised some minor remarks or suggestions on further improvement of your manuscript. You may implement these changes at the proofs editing stage if you find it necessary.

Thank you for all the hard work you have put into this. Your paper makes a strong contribution to the literature and I look forward to seeing it published.

Reviewer 1 ·

Basic reporting

Already evaluated in the original version of the manuscript.

Experimental design

Already evaluated in the original version of the manuscript.

Validity of the findings

Already evaluated in the original version of the manuscript.

Additional comments

I read the rebuttal letter and the revised version of the manuscript. I believe the Authors fully grasped that my evaluation was aimed primarily at the technical aspects of the problem, with minor suggestions regarding style and etiquete vis-à-vis potentially competing groups. The formulation of the technical problem is now straightforward. The work is an important contribution and in my view should published as is.

Reviewer 2 ·

Basic reporting

nothing to add

Experimental design

nothing to add

Validity of the findings

nothing to add

Additional comments

I believe that have had a misunderstanding throughout my comments/review and I apologize if I seemed rough, but I did not mean to be. I was hoping to improve the manuscript but I suppose that in some cases I have chosen the wrong words. I still believe in all my criticism but I could rewrite it in other words.

The manuscript had a great improvement, especially in the introduction, having merit and should be published.

I only have three more suggestions.

1) Until now, according to the bibliography that I know there would be no evolutionary sense in pursuing a phylogenetic analysis, forcing the monophyly of one group previously recovered as polyphyletic by previously analyses including three or more genes (eg Clemente-Carvalho et al. 2011; Condez et al. 2020). Such analysis would neither prove the existence of a species being recovered as monophyletic or not, nor as to the phylogenetic distance between them since such species do not have a phylogenetic relationship. I also understand that assuming that the rate of evolution in the different lineages is similar, the root at the midpoint of the path joins the two most dissimilar. However, since Brachycephalus hermogenesi and B. sulfuratus are not related, there is no possibility of this occurring. Furthermore, it still the possibility of Brachycephalus hermogenesi (municipality of Piedade) group with any other species of Brachycephalus, and a broader analysis will let this clear. I suggest you include more specimens of other species of the genus. Possibly the specimen originally identified as
B. hermogenesi from the Municipality of Piedade will be also clustered within B. sulfuratus lineage, however, I believe that you going to have more robustness, reinforce your arguments, and results in lower criticism.
Also, if you want to include all specimens of B. didactylus group, it is a 16S of B. pulex available on Genbank (MK697385).

2) The phylogenetic analysis was not included in the abstract, I suggest you include it there.

3) I was wondering if, in line 418, when you discuss the geographical discontinuity between B. hermogenesi and B. sulfuratus, if there are any examples including anuran species. I would suggest you describe one. I also believe that the following might fit:
Sabbag, Lyra, Zamudio, Haddad, Feio, Leite, Gasparini, Brasileiro. 2018. Molecular phylogeny of Neotropical rock frogs reveals a long history of vicariant diversification in the Atlantic forest, Molecular Phylogenetics and Evolution, 122, 142-156.